# Hidden Treasures of Colombia’s Pacific Mangrove: New Fungal Species and Records of Macrofungi (Basidiomycota)

**DOI:** 10.3390/jof11060459

**Published:** 2025-06-17

**Authors:** Viviana Motato-Vásquez, Lina Katherine Vinasco-Diaz, Jorge M. Londoño-Caicedo, Ana C. Bolaños-Rojas

**Affiliations:** 1Grupo de Investigación en Bilogía de Plantas y Microorganismos (BPM), Departamento de Biología, Facultad de Ciencias Naturales y Exactas, Universidad del Valle, Calle 13 No 100-00, Cali 760032, Colombia; 2Instituto de Ciencias Naturales, Universidad Nacional de Colombia, Sede Bogotá, Bogotá 111321, Colombia; vinasco.lina@correounivalle.edu.co (L.K.V.-D.); jomlondonoca@unal.edu.co (J.M.L.-C.)

**Keywords:** Colombian funga, mangrove ecosystem, new records, new species, phylogeny, taxonomy

## Abstract

Mangrove-associated fungi represent a diverse but understudied group of eukaryotic organisms, especially in the Neotropics. The Colombian Pacific region, with approximately 1300 km of coastline covered with 194,880 ha of mangrove forests that remain largely unexplored for macrofungal diversity, is recognized as a global biodiversity hotspot. This study aimed to catalog the macrofungi associated with mangrove ecosystems in Colombia, integrating morphological characterization and molecular phylogenetics, focusing on three Valle del Cauca Pacific coast localities. A total of 81 specimens were collected from both living trees and decaying wood. Detailed macroscopic and microscopic analyses were conducted, and DNA sequences from two ribosomal DNA barcode regions (ITS and LSU) were generated for 43 specimens. Three new species—*Neohypochnicium manglarense*, *Phlebiopsis colombiana*, and *Porogramme bononiae*—were documented. In addition, eight species were reported as new records for both Colombia and mangrove ecosystems, including *Microporus affinis*,* Paramarasmius palmivorus*, *Phlebiopsis flavidoalba*, *Porogramme brasiliensis*,* Resinicium grandisporum*,* Trametes ellipsospora*, *T*. *menziesii*, and *T*. *polyzona*. Although previously recorded in Colombian terrestrial ecosystems, *Lentinus scleropus* and *Oudemansiella platensis* are globally reported here for the first time from mangrove habitats. Furthermore, *Fomitopsis nivosella* and *Punctularia strigosozonata* were documented for the first time in Colombia. This study addresses the first exploration of mangrove-associated macrofungi in the country and provides new insights into the hidden fungal diversity and potential of mangrove ecosystems as a latent niche for basidiomycete dispersal along Colombia’s Pacific coast.

## 1. Introduction

Mangroves are wetland forests located in the intertidal zones of tropical and subtropical regions, such as estuaries, backwaters, deltas, creeks, and lagoons, where freshwater and saltwater meet. In terms of productivity and sustained tertiary yield, mangroves are the second most important ecosystem, following coral reefs [1,2].

While tropical forests typically harbor a high diversity of plant species, mangroves are relatively low-diversity ecosystems, with approximately 70 plant species distributed across the world’s tropical and subtropical regions [1,3,4]. The highest diversity of mangrove species is found along the Pacific and Indian Ocean coasts, particularly in Southeast Asia, which hosts the oldest and most diverse mangrove ecosystem [5]. While much is known about the diversity of animals and plants in mangroves, our understanding of fungal diversity in these ecosystems remains limited [6].

The study of mangrove fungi dates back to the 1920s, with early research by Stevens [7] and the subsequent contributions of Cribb and Cribb [8] in Australia. However, most of the early fungal records were incidental and not part of focused studies on the mangrove mycobiota [9]. Research has primarily focused on marine fungi, commonly found in estuarine or marine habitats, while terrestrial fungi in mangrove forests have been less extensively studied [10].

Macrofungi in forest ecosystems are essential for nutrient cycling and have significant roles in food webs, plant growth promotion, and medicine. Several macrofungi species are economically important as food sources, nutraceuticals, and pharmaceuticals, as well as symbiotic ectomycorrhizal fungi [11]. In mangrove ecosystems, the diversity of xylophilous basidiomycetes has been a primary focus, with taxonomic studies of corticioid fungi [12,13] and ecological studies of polypores [14,15] being prominent. A remarkable number of species are recorded from Northern Brazil, where extensive research has been conducted [16,17,18,19].

Fungal diversity in mangroves is influenced by various factors, including wood type, chemical composition, bark presence, submersion time, fungal competition, salinity, and geographical location [20]. Studies have shown that the random sampling of driftwood supports greater fungal diversity than sampling submerged woody material [21]. For example, surveys in Singapore’s Mandai mangrove revealed 41 fungal species on 188 random driftwood samples compared to 21 species on *Avicennia alba* Blume (68 samples) and 24 species on *Bruguiera cylindrica* (L.) Blume (60 twigs) [22,23].

Surveys of mangrove fungi across the Atlantic, Pacific, and Indian Oceans have identified a core group of mangrove fungi, with species being largely unique to mangrove habitats, depending on locality-specific factors. Mangrove regions with the highest species diversity include Brazil (55 species), Micronesia (19), Japan (17), and Puerto Rico (15), while other regions report fewer species [24].

Colombian Pacific mangroves extend up to 20 km inland and cover an area of 194,880 ha within the total 285,040,954 ha of mangrove forest worldwide [25]. These mangroves are classified as Very Humid Tropical Forests (bmh-T), with temperatures exceeding 24 °C and annual precipitation averaging 8000 mm [26]. The dominant species in these forests include *Avicennia germinans *(L.) L., *Conocarpus erectu*s L., *Laguncularia racemosa *(L.) Gaertn., *Mora oleífera *(Hemsl.) Ducke., *Pelliciera rhizophorae *Planch. & Triana., *Rhizophora mangle* L., *R. harrisonii *Bleech., and *R. racemosa *G. Mey [27].

The primary objective of this study was to catalog the diversity of macrofungi in Colombian mangrove ecosystems. This was achieved through both morphological and molecular analyses to identify the specimens collected. By documenting this fungal diversity, the research aims to bridge the knowledge gap regarding fungi in mangrove ecosystems and highlight their potential ecological and economic importance.

## 2. Materials and Methods

### 2.1. Sampling Area and Morphological Characterization

Fungal diversity was explored in the intertidal zone of the mangrove ecosystem along the Pacific coastline of western Colombia. The study was focused on three areas, including both primary and secondary mangrove forests. Sampling locations were mapped at a 30 s resolution in R v.4.4.1 using geospatial data and the *raster* package v.3.6-32 [28], along with visualization tools such as *ggplot2* v.3.5.1 [29] and cowplot v.1.1.3 [30] (Figure 1). A total of 81 fungal basidiomes were collected from decayed wood, twigs, and driftwood, and occasionally from living mangrove trees (*A. germinans* and *R. mangle*). All samples were carefully cataloged for subsequent laboratory analysis.

Basidiomes were photographed in situ before being removed and placed in paper bags for transport to the laboratory. Morphological characterization included a general sample inspection, focusing on macro- and micromorphological traits.

For micromorphological analysis, free-hand sections of the basidiomes were prepared on microscope slides, using 3% potassium hydroxide (KOH), Red Congo, water, or cotton blue stains. Melzer’s Reagent (IKI) was used to detect amyloid or dextrinoid reactions. Measurements were taken under 1000× magnification using cotton blue, with an accuracy of 0.1 μm. At least 25 basidiospores were measured per specimen, and the size data (with standard deviation, SD) were recorded, providing both minimum and maximum average values. Additionally, the quotient (Q) for each basidiospore was calculated and presented together with the corresponding average. The length and width of basidia hymenial cystidia and cystidioles were also recorded. Voucher specimens were deposited in the herbarium CUVC of the Universidad del Valle (Cali, Colombia).

### 2.2. DNA Extraction, Barcode Loci Amplification, and Sequencing

Total DNA was extracted directly from fresh basidiomes, dry specimens, or mycelial cultures previously ground in liquid nitrogen. DNA extraction followed the protocol outlined in the E.Z.N.A. Forensic DNA Extraction Kit (Omega Bio-Tek, Norcross, GA, USA). After quantification, DNA concentrations were normalized to 10 ng/µL for downstream molecular applications.

Two nuclear genomic loci were amplified, including the commonly used ribosomal regions ITS and LSU. The ITS rDNA region was amplified using the primer pair ITS5 (5′-GGAAGTAAAAGTCGTAACAAGG-3′) and ITS4 (5′-TCCTCCGCTTATTGATATGC-3′) [31], which generates an amplicon of approximately 800 bp. For the nuclear ribosomal Large Subunit (nLSU), the primers CTB6 (5′-GCATATCAATAAGCGGAGG-3′) [32] and LR5 (5′-TCCTGAGGGAAACTTCG-3′) [33] were used to amplify a fragment of approximately 1 kb.

PCR reactions were carried out in a final volume of 25 µL with the following composition: 15.58 µL of H_2_O, 2 µL of PCR Buffer (NH_4_)_2_SO_4_ 10×, 2 µL of MgCl_2_ 25 mM, 0.3 µL of dNTP 40 mM mix, 0.5 µL of each forward and reverse primer (10 pmol), 0.5 µL of recombinant BSA 20 mg/mL, 3 µL of genomic DNA 10 ng/µL, and 0.12 µL of Taq DNA polymerase 5 U/µL. The thermal cycling conditions were as follows: an initial denaturation at 95 °C for 5 min, followed by 36 cycles at 95 °C for 1 min, 52 °C for 45 s, and 72 °C for 1 min, with a final extension of 72 °C for 10 min. PCR products were visualized on a 1.5% agarose gel stained with GelRed^®^ to confirm amplification. Amplicons were purified following the protocol proposed by Schmitz and Riesner [34].

DNA sequencing was performed using the Sanger method with the BigDye Terminator v.3.1 Cycle Sequencing Kit (Applied Biosystems, Waltham, MA, USA). Sequencing was carried out in both directions (forward and reverse) with the same primers used as in the initial PCR. Raw sequences were assembled and manually edited using Geneious R8 software [35]. PCR primers were removed from the contig-derived consensus DNA sequences. The curated DNA sequences were deposited in the GenBank NCBI database.

### 2.3. Phylogenetic Inference

To assess the sequence similarity, we prepared datasets using the ITS and nLSU regions and compared them through BLAST searches (https://blast.ncbi.nlm.nih.gov/Blast.cgi, accessed on 14 January 2025) in the NCBI database. The sequences selected for phylogenetic analyses are listed in Table 1. Sequence alignment was performed in MAFFT v.7 [36], with the E-INS-i strategy employing no gap opening cost and equal cost for transformation (http://mafft.cbrc.jp/alignment/server/, accessed on 14 January 2025) [37,38]. Alignments were manually refined using AliView v.1.27 [39]. Initially, the ITS and nLSU sequences were aligned separately and later combined using Mesquite v.3.51 [40].

Bayesian inference (BI) analyses were conducted using MrBayes v.3.2.6 [41]. For each gene fragment (ITS and nLSU), nucleotide substitution models were chosen based on the corrected Akaike Information Criterion (AICc), calculated with jModeltest v.2.1.4 [42]. The BI analysis consisted of four MCMC runs for ten million generations, with sampling every 1000 generations. The first 25% of the sampled trees were discarded as a burn-in, and the remaining trees were used to reconstruct a 50% majority-rule consensus tree. Bayesian Posterior Probabilities (BPPs) were calculated to assess clade support, with values above 0.9 considered strongly supported.

Additionally, Maximum Likelihood analyses were conducted using IQTREE v.2.0 [43]. The optimal partition scheme and substitution models were selected using ModelFinder [44]. Ten independent runs were performed, including calculations of the Shimodaira–Hasegawa approximate likelihood ratio test (SH aLRT) [45]. The following command line was used for the analysis: iqtree -s concat.nex -spp partition.nex. best_scheme.nex -B 1000 -alrt 1000 -pers 0.2 -nstop 1000.

## 3. Results

### 3.1. Molecular Phylogeny

A total of 81 DNA sequences were generated in this study, comprising 43 for the Internal Transcribed Spacer (ITS) and 38 for the Large Subunit (LSU) ribosomal locus. From the 22 species recorded, 17 species were tested to determine their phylogenetic position. We provide detailed morphological descriptions and phylogenetic inferences for three newly described species and ten new macrofungal records in mangrove ecosystems. These additions increase the total number of species described from Neotropical mangroves to five [24,46]. All fungal isolate sequences were concatenated into datasets and analyzed using Bayesian inference and Maximum Likelihood analysis. These analyses produced congruent tree topologies with strong support; therefore, only the topologies derived from the BI analyses are presented here.

#### 3.1.1. *Paramarasmius* Phylogeny Based on Combined ITS+nLSU Sequence Data

The aligned dataset included 14 specimens representing seven species, comprising a total of 1626 characters, with 382 unique sites, 139 parsimony-informative sites, and 1285 constant sites. *Marasmius guyanensis *was chosen as the outgroup. The best-fit models estimated for Bayesian and Maximum Likelihood analyses, determined for each dataset, were HKY+F+G4 for ITS and TN+F for nLSU. The optimal tree inferred under the ML framework, based on independent runs, had a log-likelihood value of −4480.7315. The combined morphological and molecular evidence supports the placement of the studied specimen within a monophyletic clade of the genus *Paramarasmius*. The specimen was confidently identified as *P. palmivorus*, with strong statistical support for this assignment (BPP = 1.0, SH = 99%, Figure 2). Further details on the description and discussion of the species are provided below.

#### 3.1.2. *Oudemansiella* Phylogeny Based on Combined ITS+nLSU Sequence Data

The aligned dataset included 25 specimens representing six species, comprising 1686 characters, with 313 unique sites, 137 parsimony-informative sites, and 1377 constant sites. *Xerula pudens* was selected as the outgroup. The best-fit models for BI and ML analysis, determined for each dataset, were HKY+F+G4 for ITS and HKY+F for nLSU. Based on independent runs, the optimal tree inferred under the ML framework had a log-likelihood value of −4235.6264. The combined morphological and molecular evidence supports the placement of the studied specimens within the monophyletic clade of the genus *Oudemansiella*. The specimens were confidently identified as *O. platensis*, with strong statistical support for this assignment (BPP = 1.0, SH = 100%, Figure 3). Further details on the description and discussion of the species are provided below.

#### 3.1.3. *Punctularia* Phylogeny Based on Combined ITS+nLSU Sequence Data

The aligned dataset included 21 specimens representing six species, comprising 1536 characters, with 245 unique sites, 124 parsimony-informative sites, and 1313 constant sites. *Punctulariopsis yunnanensis* was selected as the outgroup. The best-fit models for BI and ML analyses, determined for each dataset, were K2P+G4 for ITS and TNe+I for nLSU. The optimal tree inferred under the ML framework, based on independent runs, had a log-likelihood value of −3660.8346. The combined morphological and molecular evidence supports the placement of the studied specimen within a monophyletic clade of the genus *Punctularia*. The specimen was confidently identified as *P. strigosozonata* with high support (BPP = 1.0, SH = 95%, Figure 4). Further details on the description and discussion of the species are provided below.

#### 3.1.4. *Resinicium* Phylogeny Based on Combined ITS+nLSU Sequence Data

The aligned dataset included 30 specimens representing 16 species, comprising 1536 characters, with 579 unique sites, 405 parsimony-informative sites, and 1030 constant sites. *Skvortzovia qilianensis* was selected as the outgroup. The best-fit models for BI and ML analyses, determined for each dataset, were TPM2u+F+I+G4 for ITS and TIM3e+G4 for nLSU. Based on independent runs, the optimal tree inferred under the ML framework had a log-likelihood value of −7017.3728. The combined morphological and molecular evidence supports the placement of the studied specimen within a monophyletic clade of the genus *Resinicium*. The specimen was confidently identified as *R. grandisporum* with high support (BPP = 1.0, SH = 100%, Figure 5). Further details on the description and discussion of the species are provided below.

#### 3.1.5. *Fomitopsis* Phylogeny Based on Combined ITS+nLSU Sequence Data

The aligned dataset included 29 specimens representing ten species, comprising 1568 characters, with 204 unique sites, 111 parsimony-informative sites, and 1419 constant sites. *Fomitopsis pinicola* was selected as the outgroup. The best-fit models for BI and ML analyses, determined for each dataset, were THKY+F+G4 for ITS and K2P+I for nLSU. Based on independent runs, the optimal tree inferred under the ML framework had a log-likelihood value of −3357.2500. The combined morphological and molecular evidence supports the placement of the studied specimen within a monophyletic clade of the genus *Punctularia*. The specimens were confidently identified as *F. nivosella *with high support (BPP = 1.0, SH = 92%, Figure 6). Further details on the description and discussion of the species are provided below.

#### 3.1.6. *Neohypochnicium* Phylogeny Based on Combined ITS+nLSU Sequence Data

The aligned dataset included 53 specimens representing 19 species, comprising a total of 1626 characters, with 512 unique sites, 345 parsimony-informative sites, and 1210 constant sites. *Abortiporus biennis* was selected as the outgroup. The best-fit models for BI and ML analyses, determined for each dataset, were TVM+F+I+R2 for ITS and TPM2+I for nLSU. The optimal tree inferred under the ML framework, based on independent runs, had a log-likelihood value of −7528.4773. The combined morphological and molecular evidence supports the placement of the studied specimen within a monophyletic clade of the genus *Neohypochnicium*. The specimens were confidently identified as a new species described as *N. manglarense* with high support (BPP = 0.90, SH = 88%, Figure 7). Further details on the description and discussion of the species are provided below.

#### 3.1.7. *Phlebiopsis* Phylogeny Based on Combined ITS+nLSU Sequence Data

The aligned dataset included 63 specimens representing 26 species, comprising 1707 characters, with 531 unique sites, 281 parsimony-informative sites, and 1305 constant sites. *Phaeophlebiopsis peniophoroides* was selected as the outgroup. The best-fit models for BI and ML analyses, determined for each dataset, were TPM2u+F+R3 for ITS and TPM3+R2 for nLSU. Based on independent runs, the optimal tree inferred under the ML framework had a log-likelihood value of −7986.1173. The combined morphological and molecular evidence supports the placement of the studied specimen within a monophyletic clade of the genus *Phlebiopsis*. Furthermore, we confirm the identity of a new species, described as *P. colombiana*, with strong support (BPP = 0.74, SH = 94%, Figure 8). This study also confirms the presence of *P. flavidoalba* in Colombia. Additionally, we present new sequences for Brazilian specimens of *P. amethystea* and *P. crassa*. The full description and discussion of the species and genus are presented below.

#### 3.1.8. *Lentinus* Phylogeny Based on Combined ITS+nLSU Sequence Data

The aligned dataset included 26 specimens representing 12 species, comprising 1525 characters, with 295 unique sites, 144 parsimony-informative sites, and 1325 constant sites. *Polyporus tricholoma* was selected as the outgroup. The best-fit models for BI and ML analyses, determined for each dataset, were K2P+R2 for ITS and TNe+I for nLSU. Based on independent runs, the optimal tree inferred under the ML framework had a log-likelihood value of −4254.7600. The combined morphological and molecular evidence supports the placement of the studied specimen within a monophyletic clade of the genus *Lentinus*. The specimens were confidently identified as *L. scleropus *with high support (BPP = 1.0, SH = 97%, Figure 9). Further details on the description and discussion of the species are provided below.

#### 3.1.9. *Microporus* Phylogeny Based on Combined ITS+nLSU Sequence Data

The aligned dataset included 14 specimens representing six species, comprising a total of 1459 characters, with 154 unique sites, 38 parsimony-informative sites, and 1298 constant sites. *Lentinus flexipes* was selected as the outgroup. The best-fit models for BI and ML analyses, determined for each dataset, were K2P+G4 for ITS and JC for nLSU. The optimal tree inferred under the ML framework, based on independent runs, had a log-likelihood value of −2966.8279. The combined morphological and molecular evidence supports the placement of the studied specimen within a monophyletic clade of the genus *Microporus*. The specimens were confidently identified as *M. affinis* (BPP = 0.73, SH = 63%, Figure 10). Further details on the description and discussion of the species are provided below.

#### 3.1.10. *Porogramme* Phylogeny Based on Combined ITS+nLSU Sequence Data

The aligned dataset included 29 specimens representing 14 species, comprising 1561 characters, with 289 unique sites, 149 parsimony-informative sites, and 1373 constant sites. *Cyanoporus aff. fuligo* was selected as the outgroup. The best-fit models for BI and ML analyses, determined for each dataset, were K2P+R2 for ITS and TNe+I for nLSU. The optimal tree inferred under the ML framework, based on independent runs, had a log-likelihood value of −3912.0072. The combined morphological and molecular evidence supports the placement of the studied specimen within a monophyletic clade of the genus *Porogramme*. Furthermore, we confirm the identity of a new species, described as *P. bononiae*, with strong support (BPP = 1.0, SH = 95%, Figure 11). This study also confirms the presence of *P. brasiliensis *in Colombia. Further details on the description and discussion of the species are provided below.

#### 3.1.11. *Trametes* Phylogeny Based on Combined ITS+nLSU Sequence Data

The aligned dataset included 42 specimens representing 17 species, comprising 1557 characters, with 420 unique sites, 260 parsimony-informative sites, and 1203 constant sites. *Hexagonia cucullata* was selected as the outgroup. The best-fit models for BI and ML analyses, determined for each dataset, were TIM3e+G4 for ITS and TNe+I+G4 for nLSU. Based on independent runs, the optimal tree inferred under the ML framework had a log-likelihood value of −6317.8400. The combined morphological and molecular evidence supports the placement of the studied specimens within a monophyletic clade of the genus *Trametes*. Furthermore, we confirm the identification of *T. menziesii* with high support (BPP = 1.0, SH = 98%); *T. sanguinea* (BPP = 1.0, SH = 99%); *T. ellipsospora* (BPP = 1.0, SH = 100%); *T. polyzona* (BPP = 1.0, SH = 100%); and *E. scabrosa* (BPP = 1.0, SH = 100%) (Figure 12). Further details on the description and discussion of the species are provided below.

### 3.2. Taxonomy

A total of 22 xylophilous basidiomycetes species have been reported from mangroves, distributed across 17 genera, 13 families, and five orders. The family Polyporaceae is the most represented, with nine species, while all other families contain only one species each. In this study, 81 specimens were collected and morphologically identified. Three new species are described. Additionally, eight species are reported as new records for both Colombia and globally for mangrove ecosystems, including *M. affinis, P. palmivorus*, *P. flavidoalba*, *P. brasiliensis, R. grandisporum, T. ellipsospora*, *T. menziesii*, and *T*. *polyzona*. *L. scleropus* and *O. platensis*, although previously recorded in Colombian terrestrial ecosystems, are reported here for the first time in the global mangrove habitats. Furthermore, *F. nivosella* and *P. strigosozonata* are documented for the first time in Colombia. The new records and the description of the new species based on morphological and molecular analyses are presented below.


**ORDER AGARICALES**

**Family Marasmiaceae**
***Paramarasmius palmivorus*** (Sharples) Antonín, Hosaka & Kolařík, Pl. Biosystems: 2 (2022).≡*Marasmius palmivorus* Sharples, Malay. agric. Journal 16 (nos. 9–10): (1928). Basionym.≡*Marasmiellus palmivorus* (Sharples) Desjardin, *Mycologia* 97(3): 670 (2005). Synonyms.Figure 2 and Figure 13A.

**Description**: Basidiomes are solitary or in small groups. The pileus is 0.4–0.7 cm in diam., convex with a small conical umbo and an involute margin, hygrophanous, and not translucently striate, or only slightly so at the margin in older specimens. Its surface is smooth at the center, becoming finely tomentose and whitish when moist, drying to a pale ochraceous–yellowish hue at the center. Lamellae are adnate, distant and cream-colored, with a concolorous, pubescent edge. The stipe is 5–7 × 1–1.2 mm, cylindrical, very finely longitudinally fibrillose, and concolorous with the lamellae. The pileipellis is a cutis composed of smooth or sparsely projecting hyphae, clamped thin- to slightly thick-walled, (3.9–)4.1–5.8(–5.9) µm in diam. Tramal hyphae are occasionally branched, smooth, clamped, thin-walled, and hyaline, IKI–, 2.9–3.9 µm in diam. Cheilocystidia are (14.7–)14.8–17.4(–17.6) × (4.9–)5.0–6.8(–6.9) µm wide, variable in form, clavate, branched, often with apical projections, thin- to slightly thick-walled, and clamped. Pleurocystidia are absent. Basidia are not observed. Basidioles are 8.8–10.8 × 2.0–2.9 μm, clavate to subfusoid, and clamped. Basidiospores are subglobose to ellipsoid, smooth, thin-walled, IKI–, and (6.4–)6.5–9.8(–10.4) × (4.6–)4.7–6.0(–6.7) μm wide; Lm = 7.9 μm, Wm = 5.2 μm, Q = 1.2–1.8 (n = 60/2).

**Distribution:** This species has been previously reported in Africa [47], India [48], Asia [49], and Brazil [50]. The present study provides the first confirmed record of the species in Colombia. Notably, this is also the first report of the species from a mangrove ecosystem.

**Specimens examined:** COLOMBIA: Valle del Cauca, Municipality of Buenaventura, Isla Quita Calzon, on decaying wood, 15 June 2023, 3°1′19.3″ N; 77°03′46.9″ W, coll. Bolaños-Rojas, A.C. et al. ACB1519 (CUVC 76329).

**Notes:** The genus *Paramarasmius* comprises agaricoid fungi characterized by a hemispherical, convex, or conical–convex pileus; a central to eccentrical stipe; smooth, thin-walled, non-dextrinoid basidiospores; well-developed cheilocystidia; and a pileipellis in the form of a cutis [51]. The species described here is a marasmioid fungus exhibiting parasitic behavior, an uncommon trait within most marasmioid lineages. It has previously been reported growing on *Elaeis guineensis* Jacq. [52,53], *Arenga engleri* Becc. [54], and *Lagerstroemia speciosa* (L.) Pers. [55], as well as on coconut palms and banana stems [47]. Until now, no species of *Paramarasmius* has been documented in Colombia. The genus currently comprises two accepted species: *P. mesosporus* and *P. palmivorus*. *Paramarasmius* differs morphologically from other related genera such as *Marasmius* Fr., *Crinipellis* Pat., and *Chaetocalathus* Singer by forming the cutis pileipellis without setiform dextrinoid hairs. The specimen described here was found growing on decaying wood. Phylogenetic analyses indicate that our specimen clusters with other specimens from Africa and Asia, forming a sister clade with *P. mesosporus*.


**Family Physalacriaceae**
***Oudemansiella platensis*** (Speg.) Speg. Anal. Soc. Cient. Argent. 12(1): 24 (1881).≡*Agaricus platensis* Speg. Anal. Soc. Cient. Argent. 9(4): 161: (1880). Basionym.≡*Psalliota platensis* (Speg.) Herter, Estudios Botánicos Región Uruguaya, III Florula Uruguayensis. Plantae Avasculares (Montevideo): 43 (1993). Synonym.Figure 3 and Figure 13B.

**Description:** Basidiomes are armillarioid. The pileus is convex, sometimes slightly depressed over the stipe, pure white, 20–40 mm broad, viscid, and radially rugulose, with small dark brown to beige pyramidal warts or patches concentrated on the disk and extending toward the limb. The margin is whitish and crenulate; the pileus pellicle is somewhat receding at the margin. Lamellae are adnate, with a decurrent tooth, slightly distant, 0.4–0.6 mm deep, and cream to white. The stipe is central, curved, cylindrical, solid, and fibrous, with a bulbous base, measuring 11–25 mm in length. The pileipellis is an ixotrichodermium composed of chains of 5–8 elongate fusiform to elongated cells, ending in a fusiform to subclavate terminal cell; cells are hyaline and thick-walled. The warts on the pileus surface are formed by a polycystoderm, with proximal cells subglobose to sphaeropedunculate, 25–30 × 23–28 μm, thin- to thick-walled, simple-septate, and hyaline with homogeneous contents. Tramal hyphae are 4–8 µm in diam., long-celled, distinctly clamped, loosely arranged, thick-walled, and hyaline, without a gelatinous matrix. Pleurocystidia are scattered, measuring (98.0–)100.9–182.2(–193.1) × (33.3–)33.5–36.9(–37.2), subglobose at the base, narrowly pedicellate, and inflated proximally, then extended into a long, broadly cylindrical neck with a rounded apex, thick-walled, and hyaline. Lamellar tramas are composed of long-celled, broadly cylindrical hyphae, constricted at the septa, 10–20 µm wide, and hyaline. Basidia are not observed. Cheilocystidia are (29.4–)30.1–35.1(–35.3) × (12.7–)12.9–14.6(–14.7) µm wide, clavate to broadly clavate, thin-walled, conspicuously clamped, occasionally with pleurocystidia, and contain scattered refractive guttules. Basidiospores are globose to subglobose, refringent, with apparently uniguiculate contents, thick-walled, hyaline, and (16.6–)16.9–21.0(–23.2) × (14.7–)15.2–20(–21.8) μm wide; Lm = 18.7 μm, Wm = 17.7 μm, Q = 1.0–1.2 (n = 90/3).

**Distribution:** The species has been recorded in Argentina, Brazil, Colombia, Costa Rica, Cuba, the Dominican Republic, and Ecuador [56,57,58].

**Specimens examined:** COLOMBIA: Valle del Cauca, Municipality of Buenaventura, Isla Quita Calzon, 3°51′19.3″ N 77°03′46.9″ W, 15 June 2023, on the decaying wood of *A. germinans* coll. Bolaños-Rojas et al. ACB1478 (CUVC7 6335); ibidem. on the decaying wood of *R. mangle*, coll. Bolaños-Rojas et al. ACB1501 (CUVC 76338); ibidem., on the decaying wood of *A. germinans*, coll. Bolaños-Rojas et al. ACB1517 (CUVC 76337); ibidem. 3°51′41.0″ N 77°03′59.2″ W, on decaying wood, 10 November 2023, coll. Bolaños-Rojas et al. ACB1571 (CUVC 76338); and ibidem. 3°51′20.2″ N 77°04′13.5″ W, 8 April 2024, coll. Bolaños-Rojas et al. ACB1669 (CUVC 76339).

**Notes:** This species has frequently been misidentified as *O. canarii* in various Latin American collections due to the latter’s long-standing status as a pantropical species. However, Corner’s [59] recent contribution questioned this premise, distinguishing *O. canarii* from *O. platensis* (South America) and describing additional species from the Old-World, including *O. lignicola* Corner, *O. crassifolia* Corner, and *O. submucida* (Corner) [57]. *Oudemansiella platensis* has been recorded in Colombia, but this is the first time the species has been recorded growing on the dead wood of mangrove species such as *A. germinans* and *R. mangle*.


**Family Pleurotaceae**
***Pleurotus djamor*** (Rumph. ex Fr.) Boedijn, Rumphius Memorial Volume: 292 (1959).≡*Agaricus djamor* Rumph. ex Fr., Syst. mycol. (Lundae) 1: 185 (1821). Basionym.=*Agaricus pacificus* Berk., London J. Bot. 1: 451 (1842). Synonym.=*Agaricus placentodes* Berk. Hooker’s J. Bot. Kew Gard. Misc. 4: 104 (1852). Synonym.Figure 13C.

**Distribution:** This species is widely documented across the Neotropics, particularly in South America. Records exist from Argentina [60], Brazil [61], Colombia [62], Mexico [63], and Puerto Rico [64], among other countries.

**Specimens examined: **COLOMBIA: Valle del Cauca, Municipality of Buenaventura, Isla Quita Calzon, 3°51′19.3″ N 77°03′46.9″ W, 15 June 2023, on the decaying wood of *R. mangle*, coll. Bolaños-Rojas, A.C. et al. ACB1482 (CUVC 76354); ibidem., on decaying wood, coll. Bolaños-Rojas, A.C. et al. ACB1518 (CUVC 76351); ibidem., on the decaying wood of *R. mangle*, coll. Bolaños-Rojas, A.C. et al. ACB1501 (CUVC 76352); and ibidem., 3°51′20.2″ N, 77°04′13.5″ W, on decaying wood, 08 April 2024, coll. Bolaños-Rojas, A.C. et al. ACB1665 (CUVC 76353).

**Notes: ***Pleurotus djamor *is a cosmopolitan species characterized by basidiomes that vary in color from white to deep salmon. The taxonomy of the pink forms of *Pleurotus* has long been debated, with at least five nominal species involved in the discussion: *P. ëous* (Berk.) Sacc., *P. flabellatus* (Berk. & Broome) Sacc., *P. ostreatoroseus* Singer, *P. salmoneostramineus* Lj. N. Vassiljeva, and *P. djamor*. Lechner et al. [60] considered all these species to be synonyms of *P. djamor*. White forms have also been reported as part of this species complex. Mating compatibility studies conducted with Brazilian specimens demonstrated interbreeding between whitish and deep salmon basidiomata, suggesting that differences in the pileus color do not justify the separation of these species [61]. The specimens collected in this study display white basidiomes and do not exhibit macroscopic or microscopic characters that differentiate them from the currently accepted concept of *P. djamor*. This species has previously been recorded degrading *R. mangle* wood in Brazil, consistent with our observations. However, further molecular and phylogenetic studies are needed to better resolve the species boundaries within this morphological variable complex.


**Family Schizophyllaceae**
***Schizophyllum commune*** Fr., Observ. mycol. (Havniae) 1: 103 (1815).Figure 13D.

**Distribution: **This species has been described in Sweden, on trunks of *Fagus*, *Alnus*, *Tilia*, *Betula*, and *Populus*. It has also been recorded in Brazil [16,19,24,65,66], Guyana [67], India [68], and Puerto Rico [64,69].

**Specimens examined: **COLOMBIA: Valle del Cauca, Municipality of Buenaventura, Isla Quita Calzon, 3°51′19.3″ N, 77°03′46.9″ W, on decaying wood, 15 June 2023, coll. Bolaños-Rojas, A.C. et al. ACB1515 (CUVC 76316); ibidem. 3°51′20.2”, N 77°04′13.5″ W, 08 April 2024, coll. Bolaños-Rojas, A.C. et al. ACB1651 (CUVC 76317); ibidem. coll. Bolaños-Rojas, A.C. et al. ACB1671 (CUVC 76318); and ibidem. coll. Bolaños-Rojas, A.C. et al. ACB1672 (CUVC 76319).

**Notes:** This species grows solitarily or gregariously as a saprotroph on decaying wood in tropical and subtropical regions, and can also colonize weakened, living wood [70]. It has been reported as a potential pathogen associated with white wood rot [71]. The species has been recorded in both mangrove ecosystems and more open, non-forested environments, suggesting ecological flexibility and adaptation to diverse substrates and moisture regimes.


**ORDER CORTICIALES**

**Family Punctulariaceae**
***Punctularia strigosozonata*** (Schweinn.) P.H.B. Talbot, Bothalia 7(1): 143 (1958).≡*Merulius strigosozonatus* Schwein., Trans. Am. Phil. Soc., New Series 4(2): 160 (1832). Basionym.≡*Stereum strigosozonatum* (Schwein.) G. Cunn., Trans. Roy. Soc. N.Z. 84: 213 (1956). Synonym.Figure 4 and Figure 13E.

**Description: **The basidiome is annual, sessile, stereoid, effused-reflexed, dimidiate to rarely corticioid, coriaceous, and up to 1.2–5.2 × 1.8–2.7 cm. The upper surface is velutinous, at first chestnut-colored, becoming concentrically sulcate with zones in varying shades of brown. It is margin chestnut when fresh, eventually dull brown and zonate, and dark brown at maturity. The context is up to 700 µm thick and heterogeneous, with an interspersed gelatinous layer. The hymenophore is phlebioid, at first smooth, developing elongate radial ridges or irregular knobs, and dark brown to violaceous. The hyphal structure is monomitic. Generative hyphae are (3.8–)3.9–4.9(–5.0) μm wide, clamped at all septa, hyaline to yellowish–brown, smooth. Tramal hyphae are hyaline, thick-walled, often embedded in a resinous layer, and (1.5–)1.5–2.0(–2.0) μm wide. Cystidia and gloeocystidia are absent. Dendrohyphidia are abundant, 20–35 × 1–2 µm, richly branched and initially hyaline, becoming yellowish to brown. Basidia are subclavate, 4-sterigmate, clamped at base, and (11.8–)11.9–13.6(–13.7) × (3.9–)3.9–4.8(–4.9) μm wide. Basidiospores are ellipsoid to subglobose, thin-walled, smooth, hyaline to yellowish, IKI–, and (4.4–)4.7–7.6 × (2.4–)2.5–4.0(–4.8) μm wide; Lm = 6.2 μm, Wm = 3.3 μm, Q = 1.6–2.4 (n = 30/1).

**Distribution: ***Punctularia strigosozonata* is primarily distributed in tropical and subtropical regions. In Europe, it is known in the eastern parts of the continent, including Estonia, Poland, Ukraine, and European Russia. In Asia, it has been recorded in China, Japan, Korea, Malaya, Taiwan, and the Asiatic part of Russia [72]. The species is also documented in South Africa, and in the South Pacific in Australia, Tasmania, and New Zealand [72]. In the Americas, it has been reported in Brazil [65,66], Canada, Chile, Venezuela, Mexico, and the United States [72].

**Specimens examined:** COLOMBIA: Valle del Cauca, Municipality of Buenaventura, 3°51′41.0″ N; 77°03′59.2″ W, on the decaying wood of *R*. *mangle*, 10 November 2023, coll. Bolaños-Rojas et al. ACB1592 (CUVC 76334).

**Notes**: *P*. *strigosozonata* typically colonizes decayed hardwoods and is associated with white rot decay [73]. Identification is relatively straightforward because of its distinctive dark coloration, reddish–brown margin, and characteristic abundance of dendrohyphidia [74]. Notably, the basidiospore measurements in the Colombian specimens are smaller than those reported from European collections by Eriksson et al. [75] and Jülich and Stalpers [76], possibly indicating regional morphological variation. Our phylogenetic analysis reveals the presence of two distinct clades within what is currently identified as *P. strigozosonata*. The Colombian specimen clusters with sequences from China, Estonia, and France, forming a well-supported clade (BPP = 0.94, SH = 95%; Figure 4). In contrast, sequences from specimens collected in the United States form a separate divergent clade with high support (BPP = 0.98, SH = 84%; Figure 4). These results suggest that *P. strigosozonata* may represent a species complex rather than a single taxon, warranting further taxonomic revision.


**ORDER DACRYMYCETALES**

**Family Dacrymycetaceae**
***Dacrymyces spathularia*** (Schwein.) Alvarenga, Mycol. Progr. 21(12, no. 96): 6 (2022).≡*Merulius spathularia* Schwein., Schr. naturf. Ges. Leipzig 1: 92 [66 of repr.] (1822). Basionym.≡*Dacryopinax spathularia* (Schwin.) G.W. Martin, Lloydia 11: 116 (1948). Synonym.Figure 13F.

**Distribution: **The species has been recorded in tropical and subtropical regions, in countries like Brazil [19], Guyana [67], Mexico [77], Puerto Rico [64], Bangladesh [78], and India [79,80].

**Specimens examined: **COLOMBIA: Valle del Cauca, Municipality of Buenaventura, Isla Quita Calzon, 3°51′20.2″ N, 77°04′13.5″ W, on decaying wood, 08 April 2024, coll. Bolaños-Rojas, A.C. et al. ACB1653 (CUVC 76350).

**Notes: **This species is easily recognizable by its orange to yellow, shiny, spathulate, and gelatinous basidiomes. It differs from other *Dacrymyces* Nees species by the pileate-stipitate basidiome, with a spathulate to flabelliform pileus, unilateral hymenium, homogeneous context, thick-walled hypha, and cylindrical abhymenial hairs and basidiospores (7.0–)8.0–10.5(–11.5) × 3.5–4.0(–4.5) µm wide. It is considered a brown rot species, typically found growing gregariously or scattered on decaying wood [81,82]. It has consistently been cited as *Dacryopinax spathularia* in all global mangrove fungal surveys.


**ORDER HYMENOCHAETALES**

**Family Rickenellaceae**
***Resinicium grandisporum*** G. Gruhn, S. Dumez & E. Schimann, Cryptog. Mycol. 38(4): 472 (2017).Figure 5 and Figure 13G.

**Description: **The basidiome is annual, widely effused, adnate, and strongly attached to the substrate. The hymenopohore is odontoid, cream-colored when fresh, and dark yellowish when dried. Aculei are 3–4 per mm, single, and have a round apex. The context is not stratified, farinaceous, white to light cream-colored when fresh, and 100–140 µm thick. The margin is gradually thinning out, minutely farinaceous, and white; hyphal cords present in the substrate and concolorous with the hymenium. The hyphal structure is monomitic. Generative hyphae with clamp connections are present at all septa, thick-walled in the subiculum, and (2.0–)2.1–3.0(–3.1) μm wide. Generative hyphae in the hymenium are agglutinated. Hyphidia are not observed. Astrocystidia are abundant in the hymenium and subiculum, usually regular and cylindrical in the central, mostly straight part, and when terminal with an enlarged base, they are covered by star-shaped crystals, and (29.4–)29.7–34.2(–34.3) × (6.9–)6.9–8.7(–8.8) μm wide. Halocystidia are not observed. Basidia are cylindrical to pyriform, with a median constriction and four sterigmata, and (19,1–)19.2–20.5(–20.6) × (6.7–)6.7–7.7(–7.8) μm wide. Basidiospores are ellipsoid, smooth, slightly thick-walled, IKI–, and (7.8–)7.8–9.8(–11) × (3.9–)3.9–6.1(–6.4) μm wide; Lm = 9.2 μm, Wm = 5.0 μm, Q = 1.4–2.5 (n = 40/1).

**Distribution:** The species has been described in French Guyana [83].

**Specimens examined: **COLOMBIA: Valle del Cauca, Municipality of Buenaventura, Isla Quita Calzon, 3°51′19.3″ N 77°03′46.9″ W, on the decaying wood of *R*. *mangle*, 15 June 23; coll. Bolaños-Rojas, A.C. et al. ACB1491 (CUVC 1348).

**Notes: **The species is well characterized by numerous astrocystidia, campled hyphae, pyriform basidia, and spores longer than 8 µm. The micro- and macromorphologies of *R. grandisporum* do not substantially differ from those of *R. bicolor*, a well-known cosmopolitan species typically found growing on Gymnosperm logs in the Northern Hemisphere. However, *R*.* bicolor* has not yet been reported in the Southern Hemisphere. The two species can be distinguished by basidiospore size: the width of *R. bicolor* is less than 3.5 µm, and the length rarely exceeds 8 µm, whereas *R. grandisporus* has longer basidiospores.

In *R. grandisporum*, astrocystidia are numerous, and halocystidia can be easily observed [83]. However, our specimen did not exhibit halocystidia. The same authors note that the presence of halocystidia can vary within specimens of *Resinicium*, and, as we collected only a single specimen, it is difficult to verify whether this characteristic is variable within and between species of the genus. Our phylogenetic analysis (Figure 5) clusters the specimen ACB1491 in a clade with other specimens identified as *R. grandisporum*, forming a sister group with *R. mutabile*. This represents the first report of *R. grandisporum* in a mangrove ecosystem and Colombia.


**Family Schizoporaceae**
***Xylodon flaviporus*** (Berk. & M.A. Curtis) Riebesehl & Langer, Mycol. Progr. 16(6): 646 (2017).≡*Poria flavipora* Berk. & M.A. Curtis ex Cooke, Grevillea 15 (no. 73): 25 (1886). Basionym.≡*Hyphodontia flavipora* (Berk. & M.A. Curtis ex Cooke) Sheng H. Wu, *Mycotaxon* 76: 54 (2000). Synonym.=*Polystictus subiculoides* Lloyd, Mycol. Writ. (Cincinnati) 7 (Letter 74): 1331 (1924). Synonym.

**Distribution: **Cosmopolitan species, recorded from South America, Africa, southern Europe, South Asia [84,85,86], Colombia [87,88], and Brazil [89,90,91,92].

**Specimens examined: **COLOMBIA: Valle del Cauca, Municipality of Buenaventura, Isla Quita Calzon, 3°51′19.3″ N 77°03′46.9″ W, on the decaying wood of *A*. *germinans*, 15 June 2023, coll. Bolaños-Rojas, A.C. et al. ACB1489 (CUVC 76332).

**Notes: **A detailed morphological examination of our specimen revealed no significant differences that would suggest a taxonomic distinction from *X. flaviporus*. However, phylogenetic analyses based on ITS sequences conducted in [93] revealed the existence of two subclades within the *X. flaviporus* lineage. Subclade 1 (FCUG 1053, KAS-GEL 3462) includes specimens from Romania and Taiwan. All additional ITS sequences of *X. flaviporus* available in GenBank originate from the Northern Hemisphere (Romania, South Korea, Taiwan, Turkey, and the USA), and consistently group within Subclade 2. Despite this genetic divergence, Riebesehl et al. [93] reported no consistent micromorphological differences between specimens from the two subclades, and the level of ITS variation observed was deemed insufficient to warrant species-level separation. Although *X. flaviporus* has been widely recorded across the Neotropics, and also reported in other mangrove ecosystems, such as *Schizopora flavipora* [90,94], no ITS sequence data are currently available from these specimens. Therefore, additional molecular and morphological studies are necessary to clarify the taxonomic identity and biogeographical structure of *X. flaviporus* in the Neotropics.


**ORDER POLYPORALES**

**Family Fomitopsidaceae**
***Fomitopsis nivosella*** (Murrill) Spirin & Vlasák, Stud. Mycol. 107: 217 (2024).≡*Tyromyces nivosellus* Murrill, *North American Flora* (New York) 9(1): 32 (1907). Basionym.≡*Polyporus nivosellus* (Murrill) Sacc. & Troter, Syll. fung. (Abellini) 21: 280 (1912). Synonym.=*Tyromyces palmarum* Murrill, *North American Flora* 9:32 (1907). Synonym.=*Polyporus durescens* Overh. ex J. Lowe, *Mycotaxon* 2(1): 65 (1975). Synonym.Figure 6 and Figure 13H.

**Description: **The basidiome is annual, sessile, or effused-reflexed, fusing in large groups, and 14–70 × 13–55 mm. Its upper surface is smooth or irregularly wrinkled, without zones, cream-colored to yellowish, and in older basidiocarps, often reddish brownish. The poroid surface is cream-colored or pale ochraceous and concave; the pores are angular, 3–5 per mm, with thin, even, or rarely serrate dissepiments. The context is rather soft, white to cream-colored, and up to 10 mm thick. Tubes are one layered, soft, concolorous with the hymenial surface, and up to 3 mm thick. The hyphal structure is dimitic; hyphae are clamped. The skeletal hyphae are hyaline or pale yellowish, (2.6–)2.9–3.9(–4.0) μm in diam. and generative hyphae are abundant to rare, hyaline, thin- to slightly thick-walled, and (2.9–)3.2–5.4(–5.9) μm in diam. Cystidioles are rare, tapering, and (12.5–)12.5–13.9(–14.0) × (4.4–)4.6–5 (–6.1) µm wide. Basidia are clavate, thin-walled, 4-sterigmate, and (10.8–)11.2–14.7 × 5.9–6.8(–6.9) μm wide. Basidioles are abundant, hyaline, thin-walled, and 12.3–13.7(–13.7) × (5.3–)5.4–8.8(–9.3) µm. Basidiospores are cylindrical to fusiform, thin-walled, hyaline, IKI–, and (4.9–)5.5–7.7(–7.9) × 2.9–3.5(–3.6) μm wide; Lm = 6.8 μm, Wm = 3.3 μm, Q = 1.6–2.4 (n = 40/3).

**Distribution: **The species has been recorded in Cuba, Brazil, Puerto Rico, and Venezuela (Spirin et al., 2024 [95]).

**Specimens examined: **COLOMBIA: Valle del Cauca, Municipality of Buenaventura, 3°51′41.0″ N 77°03′59.2″ W, on decaying wood, 10 November 2023, coll. Bolaños-Rojas, A.C. et al., ACB1573 (CUVC 76321); ibidem. ACB1580 (CUVC 76322); ibidem. ACB1587 (CUVC 76323); ibidem. 3°51′20.2″ N 77°04′13.5″ W, 08 April 2024, coll. Bolaños-Rojas, A.C. et al. ACB1654 (CUVC 76324); ibidem. coll. Bolaños-Rojas, A.C. et al. ACB1656 (CUVC 76325); ibidem. coll. Bolaños-Rojas, A.C. et al. ACB1660 (CUVC 76326); and ibidem. coll. Bolaños-Rojas, A.C. et al. ACB1675 (CUVC 76327).

**Notes: ***Fomitopsis nivosella* is distributed in North and South America, although it is rarer than its look-alike, *F. marianii *(Bres.) Spirit, Vlasák & Cartabia, which is widely distributed in the temperate forests of Eurasia, and seems to not be rare in temperate, subtropical areas of the United States [95]. In North America, it was often mixed up with other similar-looking species, particularly with *F*. *marianii*. *Fomitopsis nivosella* can be differentiated from *F*. *marianii* due to slightly smaller and more regular pores, with mostly entire orifices, as well as more abundant tramal skeletal hyphae reaching dissepiment edges, which are monomitic in *F*. *marianii*. The name *Fomitopsis nivosa* was misapplied in North and South America either to *F*. *marianii* or to *F*. *nivosella*. In Colombia, there are records of specimens identified as *Fomitopsis nivosa* (Berk.) Gilb. & Ryvarden; however, recently, Spirit et al. [95] excluded *F*. *nivosa* from the genus due to the lack of data that support its identity and phylogenetic position. Thus, the question remains as to whether the specimens cited in Colombia as *F*. *nivosa* correspond to *Polyporus nivosu*s Berk. or to *F*. *nivosella*. This is the first study in Colombia to support the record o*f F*. *nivosella* based on morphological and phylogenetic data.


**Family Neohypochniciaceae**
***Neohypochnicium manglarense*** Motato-Vásq. & Bolaños-Rojas sp. nov.Mycobank No: 858653.Figure 7 and Figure 14.

**Diagnosis: ***Neohypochnicium manglarense* is distinguished by its pale yellow to grayish yellow hymenophore that becomes cracked when drying, and by the presence of abundant subcylindrical cystidia. The basidiospores are ellipsoid, thick-walled, cyanophilous, and bear a small apiculus. The species grows on the decaying wood of *R. mangle* in mangrove ecosystems.

**Holotype: **COLOMBIA: Valle del Cauca, Municipality of Buenaventura, Isla Quita Calzon, 3°51′19.3″ N 77°03′46.9″ W, on the decaying wood of *R. mangle*, 15 June 2023, coll. Bolaños-Rojas, A.C. ACB1490 (CUVC 76331, holotype).

**Etymology: ***Manglarense* (Lat., neuter adjective), referring to the mangrove ecosystem (*manglar*) in which the species was discovered.

**Description: **The basidiome is resupinate, effuse, loosely adnate, thin, and furfuraceous to membranaceous. The hymenophore is finely reticulate (hypochnoid) and pale yellow to grayish yellow, becoming cracked upon drying. The margin is indeterminate. The subiculum has an open texture, composed of densely interwoven subhymenial hyphae. The hyphal structure is monomitic. Generative hyphae have clamps, are ramified, (2.5–)3.0–3.8(–4.0) µm wide, and thin-walled, and have slightly thickened walls at the base. Cystidia are abundant, subcylindrical, thin-walled, and measuring 25–36 × (4.0–)5.0–6.0 µm. Basidia are claviform, sometimes pedunculate, and somewhat sinuous, 10–16 × 4.5–5.5(–6.0) µm, with four sterigmata and a basal clamp. Basidiospores are ellipsoid, thick-walled, smooth, and cyanophilous, IKI–, CB–, with a small apiculus, 4.5–5.0 × (3.0–)3.1–3.6 µm; Lm = 4.8 μm, Wm = 3.5 μm, Q = 1.3–1.6(–1.7) (n = 40/1).

**Distribution: **Currently known only from the Colombian Pacific coast, where it inhabits the decaying wood of *R. mangle* in mangrove forests.

**Notes: **Morphologically, *N. manglarense* resembles *N*. *geogenium *and* N. michelii*, which also possess ellipsoid basidiospores. However, a significant difference is observed in the size of their basidiospores. *Neohypochnicium michelii* (MA- Fungi 79155, holotype) has basidiospores measuring (7.5–)9–11 × (6–)7–7.5 µm (mean = 9.4 × 7.0 µm), while *N. geogenium* (MA-Fungi 48308) has basidiospores measuring 6–9 × (4–)5–6 µm (mean = 7.9 × 5.9 µm). Additionally, *N. manglarense* may be compared to *N*. *subrigescens*, but the latter is characterized by globose to subglobose basidiospores [96], which clearly distinguishes it from the new species.


**Family Panaceae**
***Cymatoderma dendriticum*** (Pers.) D.A. Reid, Kew Bull. (13)(3): 523, 1959 [1958].≡*Thelephora dendritica* Pers., in Gaudichaud-Beaupré in Freycinet, Voy, Uranie., Bot. 4: 176 (1827) [1826,1827,1828,1829,1830]. Basionym.≡*Cladoderris dendritica* (Pers.) Berk., London J. Bot. 1(3): 152 (1842). Synonym.=*Cladoderris australis* Kalchbr. Symb. mycol. austr. 61: 442 (1878). Synonym.Figure 15A.

**Distribution: **Peninsular Malaysia [97] and Brazil [16,19,98].

**Specimens examined: **COLOMBIA: Valle del Cauca, Municipality of Buenaventura, Isla Quita Calzon, on decaying wood *R*. *mangle*, 15 June 2023, 3°51′19.3” N, 77°03′46.9” W; coll. Bolaños-Rojas, A.C. et al., ACB1480 (CUVC 76312); ibidem., ACB1485 (CUVC 76313); and ibidem., ACB1488 (CUVC 76314).

**Notes: **The species grows on dead wood (stumps, trunks, and fallen branches) and is characterized by the complex system of ridges on the lower side of the basidiome [74].


**Family Phanerochaetaceae**
***Phlebiopsis flavidoalba*** (Cooke) Hjortstam, Windahlia 17: 58 (1987).≡*Peniophora falvidoalba* Cooke, Grevillea 8 (no. 45): 21 (1879). Basionym.≡*Phanerochaete flavidoalba* (Cooke) S.S. Rattan, Biblthca Mycol. 60: 262 (1977).≡*Phlebia flavidoalba* (Cooke) Maleçon & Bertault, Acta Phytotax. Barcino. 11: 27 (1972). Synonyms.Figure 8 and Figure 16A,B,F.

**Description:** The basidiome is resupinate, effused, adnate, and up to 600 µm thick in sections. The hymenial surface is smooth to tuberculate, white when fresh, becoming beige upon drying, and occasionally sparsely and deeply cracked with age. Sterile margins are distinct, white, and pruinose. The hyphal structure is monomitic. Generative hyphae bear simple septa and are hyaline, thick-walled, interwoven, and oriented parallel to the substrate, measuring (5.9–)5.9–6.9(–6.9) μm in diam., IKI–, CB–, and unchanged in KOH. Subhymenium hyphae are agglutinated with resinous layers. Lamprocystidia are conical, hyaline, heavily covered with crystals, thick-walled, and (48.0–)51.2–79.9(–80.4) × (7.4–)7.6–12.4(–13.7) μm wide. Basidioles are clavate, thin-walled, and (16.7–)17.1–24.1(–24.5) × (5.9–)5.9–6.9(–6.9) μm wide. Basidia are clavate, thin-walled, with four sterigmata and a basal simple septum, and (16.7–)16.8–18.5(–18.6) × (3.9–)4.1–5.9(–5.9) μm wide. Basidiospores are broadly ellipsoid, hyaline, thin-walled, smooth, IKI–, CB–, and (4.9–)4.9–6.4(–6.9) × (–2.9)2.9–4.7(–4.9) μm wide; Lm = 5.5 μm, Wm = 3.9 μm, Q = 1.3–1.8 (n = 40/1).

**Distribution:** Originally described in the United States (Georgia), this species has also been recorded in Argentina [99], Brazil [100,101], Cuba [100], Hawaii [102], Guadeloupe, Uruguay [103], Venezuela [104], India [105], and Taiwan [105].

**Specimens examined: **COLOMBIA: Valle del Cauca, Municipality of Buenaventura, Isla Quita Calzon, 3°51′19.3″ N 77°03′46.9″ W; on decaying wood, 15 June 2023; coll. Bolaños-Rojas, A.C. et al. ACB1495 (CUVC 76349).

**Notes: **This species is distributed across both North and South America. It is morphologically similar to *P*. *ravenelii*, particularly in basidiospore size [106], and is associated with white rot [103,107]. This record represents the first report of the species in Colombia and the first time it has been documented in a mangrove ecosystem.

***Phlebiopsis colombiana*** Motato-Vásq. & Bolaños-Rojas sp. nov.MycoBank No: 858655.Figure 8 and Figure 16D,E.

**Diagnosis: ***Phlebiopsis colombiana* is characterized by its effused basidiome, which is smooth to tuberculate and white when fresh to beige upon drying, and its presence of conspicuous conical lamprocystidia. It is distinguished from other *Phlebiopsis* species by its notably small basidiospores, measuring 2.0–3.0 × (1.5–)1.6–2.1(–2.2) μm. This species is recorded from the Neotropical region.

**Holotype: **COLOMBIA: Valle del Cauca, Municipality of Buenaventura, Isla Quita Calzon, 3°51′19.3″ N 77°03′46.9″ W, on decaying wood, 15 June 2023, coll. Bolaños-Rojas, A.C. et al. ACB1508 (CUVC 76348, holotype).

**Etymology: **Colombiana (Lat., feminine) referring to the country where the type specimen was collected.

**Description: **The basidiome is resupinate, effused, adnate, with reflexed and incurved margins with age, and up to 400 µm thick in sections. The hymenophore is smooth to tuberculate, white when fresh, turning beige upon drying, and sometimes sparsely and deeply cracked with age. Sterile margins are distinct, white, and pruinose. The hyphal structure is monomitic. Generative hyphae bear simple septa and are hyaline, interwoven, thick-walled, (3.0–)3.2–3.5(–4.0) µm in diam., and oriented parallel to the substrate, IKI–, CB–, tissues unchanged in KOH. Subhymenial hyphae are agglutinated with resinous layers. Lamprocystidia are conical, hyaline, thick-walled (up to 5.0 µm wide), heavily encrusted with crystals, and (52–)53.4–97.1(–98) × (7.5–)7.7–19.6(–20) μm wide. Basidioles are clavate, thin-walled, abundant, and similar to basidia but slightly smaller. Basidia are clavate, hyaline, thin-walled, with four sterigmata and a basal simple septum, and 16.5–24 × 6.0–7.5 μm. Basidiospores are ellipsoid, thin-walled, hyaline, smooth, IKI–, CB–, and 2.0–3.0 × (1.5–)1.6–2.1(–2.2) μm wide; Lm = 2.6 μm, Wm = 2.0 μm, Q = 1.0–1.5(–2.0) (n = 95/4).

**Distribution: **Xylophilous occurs on decaying wood in mangrove forests of the Colombian Pacific and the Brazilian Atlantic Rainforest of Brazil.

**Additional Specimens examined: **BRAZIL: São Paulo, Parque Estadual da Serra do Mar, Trilha do Garcez, 08 May 2015, on decaying wood, Motato-Vásquez, V. et al., MV396 (SP466859, CCIBt4372). Ibidem. Parque Estadual da Cantareira, 25 February 2016, Motato-Vásquez, V. et al., MV650 (SP467025, CCIBt4563). COLOMBIA: Valle del Cauca, Municipality of Buenaventura, Isla Quita Calzon, 3°51′19.3″ N 77°03′46.9″ W, on decaying wood, 15 June 2023, coll. Bolaños-Rojas, A.C. et al., ACB1508 (CUVC).

**Notes: ***Phlebiopsis colombiana* is characterized by its effused basidiome, smooth to tuberculate hymenophore, prominent conical lamprocystidia, and small basidiospores. It is morphologically similar to and phylogenetically closely related to *P. flavidoalba*, which differs by its smooth hymenophore, larger basidiospores that are (4.9–)4.9–6.4(–6.9) × (–2.9)2.9–4.7(–4.9) μm wide, and broader distribution across North and South America [104].


**Family Polyporaceae**
***Earliella scabrosa* (Pers.)** Gilb. & Ryvarden, *Mycotaxon* 22(2): 364 (1985).≡*Polyporus scabrosus* Pers., in Gaudichaud-Beaupré in Freycinet, Voy. Uranie., Bot. (Paris) 4: 172 (1827) [1826,1827,1828,1829,1830]. Basionym.=*Daedalea conchata* Bres., Bull. Soc. mycol. Fr. 6(1): 166 (1854). Synonym.=*Earliella cubensis* Murrill, Bull. Torrey bot. Club 32(9): 479 (1905). Synonym.Figure 12 and Figure 15B.

**Distribution:** The species has been recorded from the Marianas (type locality) and Brazil [16,18].

**Specimens examined: **COLOMBIA: Valle del Cauca, Municipality of Buenaventura, Isla Quita Calzon, 3°51′20.2″ N 77°04′13.5″ W, on decaying wood, 08 April 2024, coll. Bolaños-Rojas, A.C. et al. ACB1678 (CUVC 76340).

**Notes: **This saprotrophic species colonizes dead wood and is morphologically distinctive due to its effused-reflexed basidiomata, reddish cuticle, and irregular, elongated to sinuous pores [108]. It is frequently encountered in exposed environments, likely due to its apparent ability to tolerate dry conditions for extended periods [109].

***Lentinus scleropus*** (Pers.) Fr., Syn. generis Lentinorum: 10 (1836).

≡*Agaricus scleropus* Pers., in Gaudichaud-Beaupré in Freycinet, Voy. Uranie., Bot. (Paris) 4: 167 (1827) [1826,1827,1828,1829,1830]. Basionym.

=*Agaricus hirtus* Fr., Linnaeae 5: 508 (1830). Synonym.

=*Lentinus paraguayensis* Speg., Anal. Soc. cient. argent. 16(6): 275 (1883). Synonym.

Figure 9 and Figure 15C.

**Description:** The basidiome is annual, occurring solitarily or in groups. The pileus is up to 40 mm in diam., convex to applanate, deeply depressed at the center, smooth, slightly glabrous, and white to cream when fresh, becoming light brown upon drying; the margin is lobate. The hymenophore is lamellate, with entire to forked lamellae, beige when fresh, and light brown when dry. The stipe is central, cylindrical, smooth, white to concolorous with the pileus, and rigid when dried, measuring 10 × 50 mm. The context is white in both the pileus and stipe and fleshy-tough to leathery. The spore print is white. The hyphal structure is dimitic. Generative hyphae have clamp connections and are hyaline, thin- to thick-walled, and (2.9–)2.9–4.7(–4.9) μm in diam. Skeletal-ligative hyphae are thick-walled, with 2–5 lateral branches, hyaline, and (2.0–)2.2–3.7(–3.9) μm in diam. Cystidia are absent. Basidioles are abundant, clavate, thin-walled, IKI–, CB–, and (13.7-)14.0–17.4(–17.6) × 3.9–4.7(–4.9) μm wide. Basidia are clavate, with four sterigmata, thin-walled, hyaline, and (18.6–)19.0–23.8(–24.5) × (4.9–)4.9–6.2(–6.4) μm wide. Basidiospores are ellipsoid to cylindrical, hyaline, thin-walled, IKI–, CB–, and 5.9–7.3(–7.8) × 2.0–3.7(–3.9) μm wide; Lm = 6.6 μm, Wm = 2.9 μm, Q = 2.0–3.3 (n = 40/3).

**Distribution: **Type locality: Brazil (Rio de Janeiro). The species has been recorded throughout the Neotropics, including Argentina, Barbados, Belize, Brazil, the British Virgin Islands, Bolivia, Chile, Colombia, Costa Rica, Cuba, the Dominican Republic, El Salvador, French Guiana, Grenada, Guadeloupe, Guatemala, Guyana, Haiti, Honduras, Jamaica, Martinique, Mexico, Paraguay, Puerto Rico, Saint Kitts and Nevis, the United States (Florida), Trinidad and Tobago, the Virgin Islands, and Venezuela [110].

**Specimens examined: **COLOMBIA: Valle del Cauca, Municipality of Buenaventura, Isla Quita Calzon, 3°51′19.3″ N 77°03′46.9″ W, on decaying wood, 15 June 23; coll. Bolaños-Rojas, A.C. et al. ACB1507 (CUVC 76333).

**Notes: ***Lentinus scleropus* is found on dead hardwoods. A distinctive feature of this species is the color change in the basidiocarp, transitioning from white when fresh to cinnamon brown upon drying. The absence of hyphal pegs and the nearly entire lamella edge are additional diagnostic characteristics [111]. This represents the first report of *L. scleropus* in a mangrove ecosystem.

***Microporus affinis ***(Blume & T. Nees) Kuntze, Revis. gen. pl. (Leipzig) 3(3): 494 (1898).

≡*Polyporus affinis* Blume & T. Nees, Nova Acta Phys.–Med. Acad. Caes. Leop.-Carol. Nat. Cur. 13: 18 (1826). Basionym.

=*Polyporus crenatus* Berk., Ann. Mag. Nat. Hist., Ser. 1 10: 372 (1843). Synonym.

=*Polyporus flabelliformis* Klotzsch, Linnae 8(4): 483 (1833). Synonym.

Figure 10.

**Description: **The basidiome is sessile to effused-reflexed. The pileus is semicircular, white to cream, flexible, and 18–34 × 21–25 mm. The margin is obtuse and entire. The pseudostipe is lateral to central and up to 40 mm long. The pileus surface is distinctly sulcate, glabrous, and beige with darker zonation. The pore surface is beige to pale yellow, with round pores, 4–5 per mm. The context is woody, homogeneous, white, and up to 4 mm thick. Tubes are concolorous with the context and up to 5 mm deep. The hyphal structure is dimitic. Generative hyphae have clamp connections and are hyaline, thin-walled, and (2.9–)2.9– 4.9(–5.4) µm wide. Skeletal-binding hyphae unbranched to moderately branched, mostly sinuous, thick-walled, yellowish to pale brown, (4.4–)4.4–5.7(–5.9) µm wide. Basidioles are clavate, hyaline, thin-walled, and (8.8–)9.2–12.4(–12.8) × (2.9–)3.2–4.8(–4.9) μm wide. Basidia are clavate, with four sterigmata, hyaline, thin-walled, (10.8–)10.8–13.0(–13.1) × 3.9–5.0(–5.1) µm. Basidiospores are globose, with a big guttule, hyaline, thin-walled, IKI–, CB–, and (3.8–)3.9–4.4 × (2.8–)2.9–3.9(–4.1) μm wide; Lm = 4.1 μm, Wm = 3.4 μm, Q = 1.0–1.4 (n = 40/1).

**Distribution: **Type locality: Java. The species has been recorded in Micronesia [15]. This represents the first report of the species from Colombia.

**Specimens examined: **COLOMBIA: Valle del Cauca, Municipality of Buenaventura, Isla Quita Calzon, 3°51′19.3″ N 77°03′46.9″ W, on decaying wood, 15 June 2023, coll. Bolaños-Rojas, A.C. et al. ACB1512 (CUVC 76330).

**Notes: **This species typically occurs on deciduous dead wood [112]. The present collection constitutes the first report of the species in a mangrove ecosystem and Colombian territory.


***Porogramme bononiae* Bolaños-Rojas & Motato-Vásq. sp. nov.**
Mycobank No: 858656.Figure 11 and Figure 17A–C,E.

**Diagnosis:** *Porogramme bononiae* is recognizable within the genus by its combination of 2–3 pores per mm, white hymenophore, presence of dendrohyphidia in the dissepiments, and cylindrical basidiospores.

**Holotype: **COLOMBIA: Valle del Cauca: Municipality of Buenaventura, Isla Quita Calzon, 3°51′19.3″ N 77°03′46.9″ W, on decaying wood, 15 June 2023, coll. Bolaños-Rojas, A.C. et al. ACB1494 (CUVC 76346, holotype).

**Etymology: ***Bononiae* (Lat., feminine), in honor of Prof. Dra. Vera Lucía Ramos Bononi (Brazil), in recognition of her significant contributions to the field of mycology and her role in mentoring new generations of mycologists in Brazil and South America.

**Description: **The basidiome is annual, resupinate, inseparable, corky when fresh, becoming brittle when dry, up to 15 cm long, 4.8 cm wide, and 0.9 mm thick. The margin is sterile and very narrow to almost absent. The pore surface is white when fresh to yellowish upon drying. Pores are angular and 2–3 per mm; dissepiments are thin and entire to slightly lacerate. The subiculum is white, corky, and up to 0.2 mm thick. Tubes are concolorous with the pore surface, up to 0.8 mm deep. The hyphal structure is dimitic. Generative hyphae have clamp connections and are hyaline, thin-walled, occasionally branched, and 2.0–3.0(–3.2) µm in diam. Skeletal hyphae are frequent, hyaline, thick-walled with a narrow lumen, moderately branched, interwoven, and (3.6–)4.0–4.5(–5.0) µm in diam. Cystidia and cystidioles are absent. Dendrohyphidia are frequent at the edges of dissepiments. Hyphal pegs are occasionally present. Basidioles are similar in shape to basidia, but slightly smaller. Basidia are clavate, with four sterigmata and a basal clamp, and (16–)19–21 × (5.5–)6.0–7.0 µm wide. Irregularly shaped crystals are frequently observed among the hymenium. Basidiospores are cylindrical, tapering at the apiculus, hyaline, thin-walled, smooth, IKI–, CB–, and (5.2–)5.5–7.2(–7.5) × 4.0–5.0 μm; Lm= 6.5 μm, Wm= 4.7 μm, Q= (1.1–)1.2–1.6(–1.7) (n=50/2).

**Additional specimens examined: **COLOMBIA: Valle del Cauca, Municipality of Buenaventura, Isla Quita Calzon, 3°51′19.3″ N 77°03′46.9″ W, on the decaying wood of *R*. *mangle*, 15 June 2023, coll. Bolaños-Rojas, A.C. et al. ACB1486 (CUVC 76345).

**Distribution:** Currently only known from the type locality, Isla Quita Calzon, Buenaventura, Colombia, where it grows on the decaying wood of *Rhizophora mangle*.

**Notes: ***Porogramme bononiae* is morphologically similar to *P. brasiliensis*, but can be distinguished by its larger pores and wider basidiospores. Additionally, *P. bononiae* has a white hymenophore when fresh, whereas *P. brasiliensis* typically displays a bluish–white coloration. Phylogenetically, *P. bononiae* forms a sister clade (BPP = 1.0, SH = 90%, Figure 11) with *P. subargentea*, a species from Brazil. The two differ morphologically, as* P. subargentea* features dextrinoid skeletal hyphae and longer, larger basidiospores [113].

***Porogramme brasiliensis ***(Ryvarden) Y.C. Dai, W.L. Mao & Yuan Yuan, IMA Fungus 14 (no. 5): 10 (2023).≡*Grammothele brasiliensis* Ryvarden, Syn. Fung. (Oslo) 33: 38 (2015). Basionym.Figure 11 and Figure 17D,F.

**Description: **The basidiome is resupinate, widely effused, strongly adnate, hard, and brittle. The margin is narrow and white. The pore surface is dark gray, with rounded and entire pores, sometimes slightly lacerated, and (6–)7–8 per mm. Dissepiments contain white, irregular crystals. Tubes are shallow, up to 210 µm deep, and grayish. The context is white and very thin to nearly invisible. The hyphal structure is dimitic. Generative hyphae have clamp connections and are hyaline, 2–4 µm wide, but difficult to observe due to heavy encrustation with crystals and a strongly agglutinated hyphae structure. Skeletal hyphae are brownish, thick-walled, strongly agglutinated in bundles, and covered with crystals, simulating large metuloid cystidia, dextrinoid in Melzer’s reagent, and 3–6 µm wide. Dendrohyphidia are present, though sometimes difficult to distinguish. Sterile hyphal ends in the hymenium simulate narrow, cylindrical cystidioles. Basidia clavate have four sterigmata and are hyaline, measuring 15–17 × 4–6 µm. Basidiospores are cylindrical, hyaline, thin-walled, smooth, IKI–, and CB–, measuring (5.0–)5.5–6.0 × 2.5–3.0 μm; with Lm= 5.5 μm, Wm =2.7 μm, and Q =1.8–1.9(–2.2) (n = 35/1).

**Distribution: **Originally described in São Paulo, Brazil [109], with approximately 70 GBIF records, all within Brazil, mainly from Amazonia and the Atlantic Forest. This finding from Buenaventura, Colombia, extends the known distribution approximately 770 km northwest, marking the first record in the Colombian Pacific mangrove ecosystem, on *A. geminans*.

**Specimens examined: **COLOMBIA: Valle del Cauca, Municipality of Buenaventura, Isla Quita Calzon, 3°51′19.3″ N 77°03′46.9″ W, on the decaying wood of *A. germinans*, 15 June 2023, coll. Bolaños-Rojas, A.C. et al. ACB1510 (CUVC 76347).

**Notes**: This species is typically found on dead hardwood and is distinguished by its dark gray hymenophore, regular round pores, and crystal-covered skeletal hyphae [109]. It may be confused with *Tinctoporellus epimiltinus* (Berk. & Br.) Ryvarden, which shares similar coloration (bluish–white and light beige) and pore dimensions (angular to round, 7–9 per mm) and a dimitic hyphal structure. However, *T. epimiltinus* differs in its ellipsoid to subglobose basidiospores [4–5(5.5) × 2.5–3.0 µm], and notably its absence of dendrohyphidia and cystidioles, which are characteristic of *P. brasiliensis* [113].

***Trametes ellipsospora*** Ryvarden, *Mycotaxon* 28(2): 539 (1987).Figure 12 and Figure 15D.

**Description: **The basidiome is annual, pileate to effused-reflexed, and usually imbricate. The pileus is dimidiate to almost circular, with a distinctly umbilicate base, flexible when fresh, projecting up to 38–40 × 35–38 mm. The pilear surface is cream to pale gray upon drying and glabrous to slightly velutinous, exhibiting prominent concentric zoning and sulcation. The margin is thin and sharp. The pore surface is cream to straw-colored with glancing luster. Pores are round to angular, 4–6 per mm; dissepiments are entire and thin. The context is white to cream, coriaceous, homogeneous, and up to 600 µm thick. The hyphal structure is trimitic. Generative hyphae are hyaline, thin-walled, with occasional clamp connections, and (1.4–)1.5–2.9–3.2) µm in diam. Skeletal hyphae are dominant, hyaline, thick-walled to solid, sparsely branched, interwoven, and (2.9–)3.0–5.9(–6.1) µm in diam. Binding hyphae are hyaline, thick-walled to almost solid, branched, densely interwoven, and 1.5–3.2 µm in diam. Cystidia are absent. Fusoid cystidioles occasionally present in the hymenium and are hyaline and thin-walled. Basidia are clavate, bearing four sterigmata with a basal clamp connection, and 9–17 × 3.7–5.0 µm wide. Basidioles are similar in shape to basidia, but slightly smaller. Basidiospores are ellipsoid, hyaline, thin-walled, smooth, IKI–, CB–, and (3.8–)3.9–4.4 × (2.5–)2.7–3.4(–3.6) µm wide; Lm = 4.1 μm, Wm = 3.0 μm, Q = 1.2–1.5 (n = 30/1).

**Distribution:** *Trametes ellipsospora* is known from the type locality at Neblima, Venezuela [114]. Additional records include Brazil [115], China [116], India [117], and the Philippines [118]. The species appears to inhabit a range of subtropical to tropical regions, typically occurring on decaying wood in coastal and lowland forest ecosystems.

**Specimens examined: **COLOMBIA: Valle del Cauca, Municipality of Buenaventura, Isla Quita Calzon, 3°51′19.3″ N 77°03′46.9″ W, on the decaying wood of *R*. *mangle*, 15 June 2023, coll. Bolaños-Rojas, A.C. et al. ACB1493 (CUVC 76344).

**Notes: ***Trametes ellipsospora* is characterized by its thin, flexible, often dimitiate to nearly circular basidiome, with a glabrous to slightly velutinous pilear surface and small, round to angular pores (4–6 per mm) [114]. This species closely resembles* T*. *marianna* (Pers.) Ryvarden, which differs in having longer, cylindrical basidiospores (6–7 × 2–2.5 µm; [119]. It also shares some macromorphological traits with *T. pavonia* (Hook.) Ryvarden, such as a velutinous pileus and similar-sized pores (5–6 per mm), but the latter species has significantly larger basidiospores (5–6 × 3–4 µm) [120]. The Colombian record represents a new addition to the known distribution of *T. ellipsospora*, expanding its range by approximately 1863.1 km northwest of the type locality.

***Trametes menziesii*** (Berk.) Ryvarden, Norw. Jl. Bot. 19(3–4): 236 (1972).≡*Polyporus menziesii* Berk., Ann. Mag. nat. Hist., Ser. 1 10: 378 (1843). Basionym.≡*Cubamyces menziesii *(Berk.) Lücking, Willdenowia 50(3): 396 (2020). Synonym.=*Trametes grisea* Pat., J. Bot., Paris 11: 341 (1897). Synonym.Figure 12 and Figure 15E.

**Description:** The basidiome is annual to perennial, pileate to effused-reflexed, and sessile. The pileus is dimidiate, flabelliform to semi-circular, glabrous or with a central protuberance in some specimens, flexible, and measures 20–105 × 25–75 mm. The upper surface is hirsute, with sulcate zones, beige to gray, and concentric zoning often accentuated near the margin. The margin is thin and usually deflexed when dry. The hymenophore has cream-colored, angular pores; 2–3 pores per mm. The context is white to cream, homogeneous, flexible, and up to 200 μm thick. The hyphal structure is trimitic. Generative hyphae have clamp connections and are hyaline, thin-walled, IKI–, and (1.4–)1.5–2.1(–2.2) μm in diam. Skeletal hyphae are hyaline, thick-walled, IKI–, and (4.7–)4.8–5.8(–6.0) μm in diam. Binding hyphae are abundant in the context and (5.4–)5.5–6.0 µm in diam., usually with twisted branches. Basidia are not observed. Basidiospores are subcylindrical to oblong, hyaline, thin-walled, often difficult to find in dry specimens, IKI–, CB–, and (2.4–)4.4–6.0(–6.1) × (1.8–)1.9–2.9(–4.0) μm wide; Lm = 5.2 μm, Wm = 2.5 μm, Q = 1.8–2.6 (n = 30/1).

**Distribution:** *Trametes menziesii* is a pantropical species widely distributed across tropical regions. It has been reported in most Sub-Saharan African countries [119,121,122]. This study presents the first confirmed record of the species in Colombia and within a mangrove ecosystem.

**Specimens examined: **COLOMBIA: Valle del Cauca, Municipality of Buenaventura, Isla Quita Calzon, 3°57′53.7″ N 77°22′27.5″ W, on decaying wood, 26 February 2023, coll. Bolaños-Rojas, A.C. et al. ACB1346 (CUVC 76373); ibidem. coll. Bolaños-Rojas, A.C. et al. ACB1349 (CUVC 76274); ibidem. coll. Bolaños-Rojas, A.C. et al. ACB1350 (CUVC 76275); ibidem. coll. Bolaños-Rojas, A.C. et al. ACB1351 (CUVC 76276); ibidem. coll. Bolaños-Rojas, A.C. et al. ACB1354 (CUVC 76277); ibidem. ACB1359 (CUVC 76278); ibidem. 4°02′20.0″ N 77°14′04.9″ W, on decaying wood, 22 April 2023, coll. Bolaños-Rojas, A.C. et al. ACB1367 (CUVC 76279); ibidem. coll. Bolaños-Rojas, A.C. et al. ACB1368 (CUVC 76280); ibidem. coll. Bolaños-Rojas, A.C. et al. ACB1374 (CUVC 76281); ibidem. coll. Bolaños-Rojas, A.C. et al. ACB1375 (CUVC 76282); ibidem. coll. Bolaños-Rojas, A.C. et al. ACB1377 (CUVC 76283); ibidem. coll. Bolaños-Rojas, A.C. et al. ACB1378 (CUVC 76284); ibidem. 3°51′19.3″ N 77°03′46.9″ W, on a living tree of *A*. *germinans*, 15 June 2023, coll. Bolaños-Rojas, A.C. et al. ACB1473 (CUVC 76285); ibidem. coll. Bolaños-Rojas, A.C. et al. ACB1484 (CUVC 76286); ibidem., on decaying wood, coll. Bolaños-Rojas, A.C. et al. ACB1496 (CUVC 76287); ibidem. coll. Bolaños-Rojas, A.C. et al. ACB1499 (CUVC 76288); ibidem. coll. Bolaños-Rojas, A.C. et al. ACB1500 (CUVC 76289); ibidem. coll. Bolaños-Rojas, A.C. et al. ACB1514 (CUVC 76290); ibidem. 3°51′41.0″ N 77°03′59.2″ W, on decaying wood, 10 November 2023, coll. Bolaños-Rojas, A.C. et al. ACB1572 (CUVC 76291); ibidem. coll. Bolaños-Rojas, A.C. et al. ACB1574 (CUVC 76292); ibidem., on decaying wood *R*. *mangle*, coll. Bolaños-Rojas, A.C. et al. ACB1576 (CUVC 76293); ibidem., on decaying wood, coll. Bolaños-Rojas, A.C. et al. ACB1583 (CUVC 76294); ibidem. coll. Bolaños-Rojas, A.C. et al. ACB1584 (CUVC 76295); ibidem. coll. Bolaños-Rojas, A.C. et al. ACB1590 (CUVC 76296); ibidem. coll. Bolaños-Rojas, A.C. et al. ACB1594 (CUVC 76297); ibidem. 3°51′20.2″ N 77°04′13.5″ W, on decaying wood, 08 April 2024, coll. Bolaños-Rojas, A.C. et al. ACB1649 (CUVC 76298); ibidem. coll. Bolaños-Rojas, A.C. et al. ACB1650 (CUVC 76299); coll. Bolaños-Rojas, A.C. et al. ACB1658 (CUVC 76300); ibidem. coll. Bolaños-Rojas, A.C. et al. ACB1661 (CUVC 76301); ibidem. ACB1662 (CUVC 76302); ibidem. coll. Bolaños-Rojas, A.C. et al. ACB1667 (CUVC 76303); and ibidem. 3°51′46.4″ N 77°04′13.7″ W, on decaying wood, 15 November 2024, coll. Bolaños-Rojas, A.C. et al. ACB1844 (CUVC 76304).

**Notes:** *Trametes menziesii* commonly occurs on dead trunks, logs, and stumps in open and dry tropical environments [119]. The species is associated with white rot decay [123]. This study presents the first documented occurrence of *T. menziesii* in Colombia and specifically in a mangrove ecosystem, suggesting a broader ecological amplitude than previously recognized.

***Trametes polyzona*** (Pers.) Justo, Taxon 60(6): 1580 (2011).≡*Polyporus polyzonus* Pers. in Gaudichaud-Beaupré in Freycinet, Voy. Uranie., Bot. (Paris) 4: 171 (1827). Basionym.≡*Coriolopsis polyzona* (Pers.) Ryvarden, Norw. Jl. Bot. 19: 230 (1972). Synonym.≡*Daedalea polyzona* (Pers.) Pers., Mycol. eur. (Erlanga) 3: 8 (1828) Synonym.Figure 12 and Figure 15F.

**Description: **The basidiome is pileate, sessile to imbricate, and flexible. The pileus surface is zonate, brownish, velutinous, and frequently colonized by green algae forming irregular patches. The basidiome measures 21–144 × 2–100 mm and is solitary or occurs in groups. The hymenophoral surface is light brown, with round to angular pores; 2–3 per mm. The context is 2–3 mm thick, heterogeneous, and has a distinct two-layered structure: a light brown lower layer and a paler upper layer, occasionally separated by a thin black line. The hyphal structure is trimitic. Generative hyaline are thin-walled, with clamp connections, IKI–, and (1.5–)1.5–2.3(–2.4) µm in diam. Skeletal hyphae are yellow–brown, thick-walled, IKI–, and (4.9–)5.0–7.1(–7.2) µm in diam. Connective hyphae are hyaline, weakly branched, and (2.9–)2.9–4.3(–4.5) µm in diam. Basidia are not observed. Basidiospores are ellipsoid to subcylindrical, smooth, hyaline, thin-walled, IKI–, CB–, and (4.9–)5.0–7.4(–7.4) × (2.5–)2.7–3.4(–3.5) µm wide; Lm = 6.1 μm, Wm = 3.0 μm, Q = 1.6–2.7 (n = 30/1).

**Distribution: **This species has been reported in Mexico [124] and Brazil [90,125]. In this study, *T. polizona* is documented for the first time in Colombia and from a mangrove ecosystem, highlighting its presence in coastal tropical forests on the decaying wood of *A. germinans* and *R. mangle*.

**Specimens examined: **COLOMBIA: Valle del Cauca, Municipality of Buenaventura, Isla Quitacalzón, 3°51′19.3″ N 77°03′46.9″ W, on the decaying wood of *A*. *germinans*, 15 June 2023, coll. Bolaños-Rojas, A.C. et al. ACB1475 (CUVC 76305); ibidem., on the decaying wood of *R*. *mangle*, coll. Bolaños-Rojas, A.C. et al. ACB1492 (CUVC 76306); ibidem. 3°51′41.0″ N 77°03′59.2″ W, on the decaying wood of *A*. *germinans*, coll. Bolaños-Rojas et al. ACB1585 (CUVC 76308); ibidem., on decaying wood, coll. Bolaños-Rojas et al. ACB1577 (CUVC 76307); ibidem. coll. Bolaños-Rojas et al. ACB1589 (CUVC 76309); ibidem. 3°51′20.2″ N 77°04′13.5″ W, on decaying wood 8 April 2024, coll. Bolaños-Rojas et al. ACB1652 (CUVC 76310); and ibidem. coll. Bolaños-Rojas et al. ACB1670 (CUVC 76311).

**Notes: **This is a saprobic species associated with the decomposition of hardwood [120]. Morphologically, it resembles other species in the genus such as *T. hirsuta*, *T. betulina*, *T. socotrana*, *T. villosa*, and *T. maxima*, particularly in the hirsute pileus with narrow sulcate zones [126].


**
*Trametes sanguinea*
**
≡*Boletus sanguineus* L. Sp. pl., Edn 2 2(2): 1646 (1763). Basionym.≡*Fabisporus sanguineus* (L.) Zmitr., Mycena 1(1): 93 (2001). Synonym.≡*Pycnoporus sanguineus* (L.) Murrill, Bull. Torrey bot. Club 31(8): 421 (1904). Synonym.Figure 12 and Figure 15G.

**Distribution:** The species has been widely recorded in different countries of South America, including Brazil and Colombia [16,18,66,98,101], Puerto Rico [64], Mexico [77], and India [68,80].

**Specimens examined: **COLOMBIA: Valle del Cauca, Municipality of Buenaventura, Isla Quita Calzon, 3°58′10.3″ N 77°22′35.3″ W, on decaying wood, 26 February 2023, coll. Bolaños-Rojas, A.C. et al. ACB1358 (CUVC 76341); ibidem. 3°51′19.3″ N 77°03′46.9″ W, on the decaying wood of *A*.* germinans*, 15 June 2023, coll. Bolaños-Rojas, A.C. et al. ACB1476 (CUVC 76342).

**Notes: **This species is characterized by its distinctive red–orange basidiome. It typically colonizes fallen or standing trunks of various hardwoods and is frequently found in open, sun-exposed environments, including fence posts and stored timber [114].

## 4. Discussion

In Colombia, the fungal diversity associated with mangrove ecosystems remains significantly underexplored, with a potential number of species yet to be described or formally published. Nevertheless, the ecological heterogeneity of western Colombia, particularly along the Pacific coast, suggests a high potential for the discovery of novel species, regional endemism, and untapped genetic resources. This area continues to represent a biological “dark spot” for fungal diversity [127].

This study represents the first assessment of mangrove-associated macrofungi in Colombia, integrating both morphological and molecular approaches. The identification of *Neohypochnicium manglarense, Phlebiopsis colombiana,* and *Porogramme bononiae* as new species, alongside ten new macrofungal records for mangrove ecosystems, underscores the importance of prioritizing this geographical region for documenting and categorizing fungal diversity. A deeper understanding of the ecological interactions and adaptive strategies of these fungal communities in such complex environments is critical to bridging the current knowledge gap in one of the world’s biodiversity hotspots. Extensive and systematic sampling is necessary to elucidate new populations and their ecological preferences, including the dispersal status (e.g., native or endemic), host specificity, and adaptability to niche conditions.

Unlike terrestrial Neotropical forests, which are characterized by high tree diversity, mangrove ecosystems are dominated by relatively few plant species, which may facilitate the establishment of fungi adapted to limited host diversity [97]. Although host specificity was not directly evaluated in this study, the dominance of *A. germinans* and *R. mangle* at the sampling sites may influence the composition and structure of the fungal community. Additionally, environmental factors such as variable salinity and periodic flooding contribute to niche differentiation and may further promote fungal diversity in mangrove ecosystems [128].

Despite Colombia’s Pacific coastline extending for approximately 1300 km and encompassing an estimated 194,880 hectares of mangrove ecosystems [25], this study explored just three locations, representing less than 0.001% of the total mangrove area. No data were collected on seasonal variation (wet–dry periods) or host specificity. Most identified species were saprophytic Basidiomycota, primarily from the Polyporales and Agaricales orders.

Numerous macrofungal species, members of Basidiomycota, have been documented in mangrove ecosystems across tropical and subtropical latitudes [8,77,86,92,109,129,130], with the composition and dominant taxa varying regionally. In the present study, 22 Basidiomycota species were identified, with *T. menziesii* emerging as the most prevalent across all sampling sites. This marks the first recorded occurrence of *T. menziesii* in Colombia and in a mangrove ecosystem. In comparison, Brazilian mangroves are commonly associated with dominant species such as *Tyromyces chioneus* (Fr:Fr) Donk and *Stereum cinerascens* [17,19,65,131]. Other species such as *T. sanguinea*, *S. commune*, and *D spathularia *are recurrently reported in mangrove studies from various regions [77,98,131].

Comparative data on Basidiomycota diversity in mangroves across different regions reveal marked variability: 55 species in Brazil [24], 32 in Bangladesh [78,132], 19 in Micronesia [15], 16 in Japan [133], 17 in Guyana [67], 14 in Puerto Rico [64] 12 in Panama [134], and 4 in Mexico [77]. Notably, the 22 species identified in the present study position Colombia as the third most diverse Neotropical country in terms of macrofungi associated with mangrove ecosystems.

This study demonstrates that the Colombian Pacific mangrove ecosystem serves as both a reservoir of novel fungal species and a potential dispersal niche for macrofungi in Basidiomycota. Phylogenetic evidence strongly supports the placement of the new taxa, and the new distribution records are well corroborated by both morphological and molecular data. For instance, *L. scleropus*, *M. affinis*, *O. platensis*, *P. palmivorus*, *P. flavidoalba*, *R. gransdisporum*, *P. brasiliensis*, *T. menziesii*, *T*. *ellipsospora*, and *T. polyzona* correspond to the first mangrove-associated species records.

The majority of phylogenetic studies on mangrove-associated fungi have been based on morphological descriptions, often limiting the resolution and accuracy of species identification. In fact, in our results, many of the analyzed species group together with species recorded in Africa, Asia, or the Northern Hemisphere. Some of these species even exhibit morphological variations compared to the description of type species, as is the case with *P. djamor*, *P. strigosozonata*, and *X. flaviporus*. The lack of molecular data from Neotropical specimens, and especially from specimens collected in mangrove ecosystems, may be obscuring a greater diversity of undescribed species.

Furthermore, the integration of the DNA of terrestrial fungal isolates into phylogenetic reconstructions may influence the identification of potential genetic imprints due to the underrepresentation of marine or estuarine lineage strains, which can obscure the potential genetic signatures unique to mangrove-adapted species. Many of the basidiomycetes previously reported in mangrove environments are related to terrestrial taxa [135,136], suggesting that these communities may be transient or facultatively adapted to complex environments, such as mangrove forests, not only through osmotic balance, but also through host colonization. However, the newly reported species and mangrove-associated records necessitate careful analysis, as there are no terrestrially documented counterparts. Consequently, further extended sampling efforts and integrative information, combining both morphological and molecular data, are required to determine the origin, distribution, and ecological roles of these fungi.

Several fungal species of Basidiomycota previously reported in mangrove forests, including *S. commune*, *T. sanguinea*, and *P. strigosozonata*, have demonstrated host specificity, particularly toward *R. mangle* and *A. schaueriana* Stapf & Leechm. ex Moldenke LC as saprotrophs [66]. In our study, the presence of *R. mangle* and *A. germinans* correlates with previous reports of fungal communities common to these mangrove species [14,77]. The relatively low plant diversity in mangrove ecosystems may enhance fungal colonization opportunities and promote dispersal within this constrained niche [14].

Finally, the discovery of edible species such as *O. platensis* and *P. djamor* along Colombia’s Pacific coast presents promising socioeconomic and ecological implications. As one of the most socioeconomically vulnerable regions, increased access to alternative nutritious food sources could positively reshape local perceptions of mangrove ecosystems. This, in turn, may foster stronger support for local conservation initiatives. With continued documentation of fungal biodiversity and increasing recognition of the multifaceted value of mangrove ecosystems, the expansion of protected areas and ecological restoration initiatives continues to gain momentum [4,137].

The findings presented here contribute to knowing the macrofungal diversity associated with mangrove ecosystems and offer promising genetic resources with potential biotechnological applications [138]. However, habitat degradation remains a pressing threat. Deforestation in the region has led to a 6.88% loss of mangrove cover over the past decade [139], significantly reducing host and substrate availability due to selective logging and the decline of old-growth forests [97]. Notably, in the sampling areas investigated here, we observed that some young mangrove forests were often devoid of macrofungi due to the absence of mature wood, twigs, or woody senescent trees. In contrast, old-growth mangrove forests with abundant decaying wood material supported greater fungal richness and basidiome development. In summary, fungal mangrove species exhibit remarkable adaptability to extreme environmental conditions. The Colombian Pacific coast represents a promising frontier for macrofungal discovery, offering both biodiversity insights and access to novel genetic resources with potential applications.

## 5. Conclusions

Through morphological characterization and molecular phylogenetics, we identified three novel species and documented ten new records for mangrove ecosystems and Colombia, elevating the country to the third most diverse Neotropical country in terms of known mangrove-associated macrofungi. Our findings underscore the ecological significance of old-growth mangrove forests as critical habitats for diverse Basidiomycota lineages, many of which lack terrestrial counterparts and may represent previously unrecognized endemic or estuarine-adapted taxa. The presence of edible and potentially biotechnologically valuable species further highlights the relevance of these ecosystems for local livelihoods and sustainable development. However, ongoing habitat degradation, driven by deforestation and the decline of mature forests, poses a significant threat to these fungal communities. This study advocates for continued sampling, long-term ecological monitoring, and the inclusion of molecular data to better understand species origins, host associations, and adaptive strategies in mangrove environments. The Colombian Pacific mangroves thus represent a promising frontier for fungal discovery, conservation, and the development of ecosystem-based solutions.

## Figures and Tables

**Figure 1 jof-11-00459-f001:**
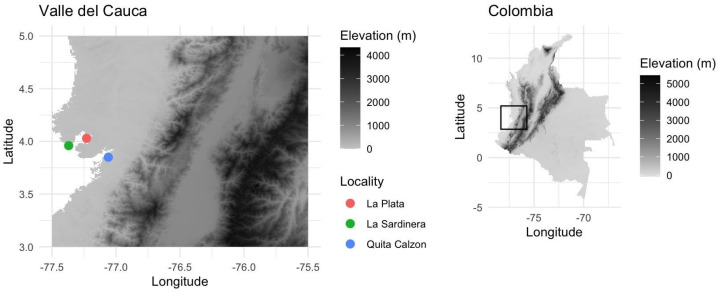
Graphical representations of the sampling area for macrofungi analyzed in this study. The map illustrates altitude (in meters above sea level) on a 30 s grid, covering Colombia and the Valle del Cauca region. Sampling localities along the Pacific coastline are color-coded.

**Figure 2 jof-11-00459-f002:**
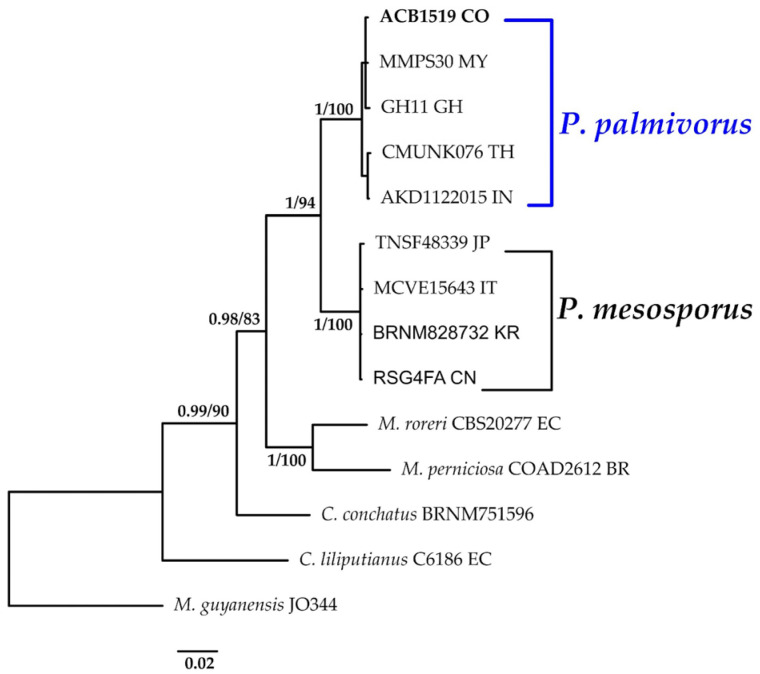
Phylogenetic relationships within the *Paramarasmius* clade inferred from a combined dataset of ITS and nLSU sequences using Bayesian Inference. Specimens sequenced in this study are indicated in bold. Values at nodes represent Bayesian Posterior Probabilities (BPPs, left) and Shimoradai–Hasegawa approximate likelihood ratio test (SH, right). Country codes following voucher specimens correspond to their country of origin. The bar indicates the expected number of substitutions per site.

**Figure 3 jof-11-00459-f003:**
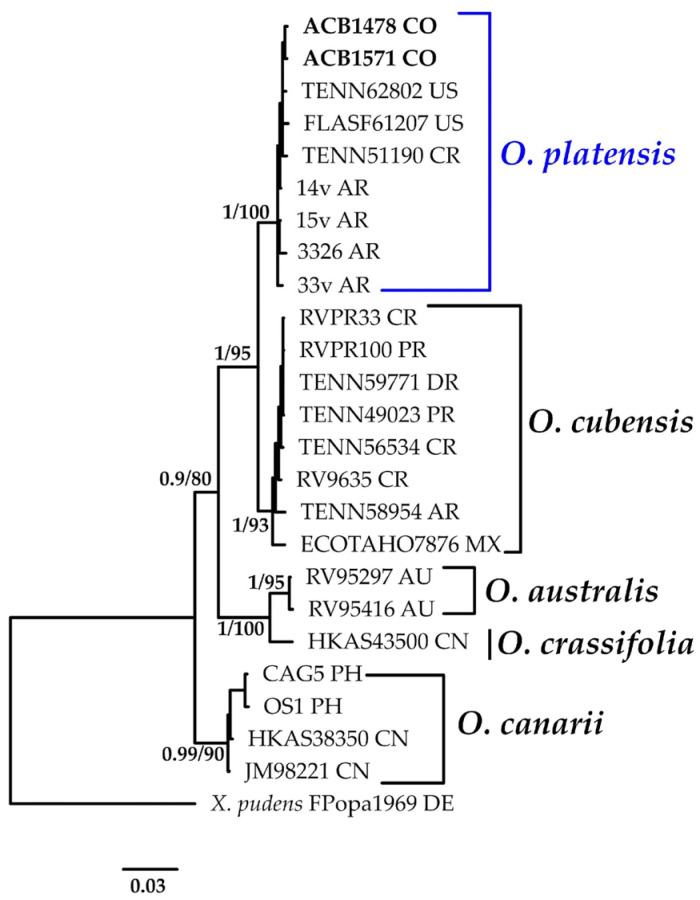
Phylogenetic relationships within the *Oudemansiella* clade inferred from a combined dataset of ITS and nLSU sequences using BI. Specimens sequenced in this study are indicated in bold. Values at nodes represent Bayesian Posterior Probabilities (BPPs, left) and Shimoradai–Hasegawa approximate likelihood ratio test (SH, right). Type specimens are marked with a (T). Country codes following voucher specimens correspond to their country of origin. The bar indicates the expected number of substitutions per site.

**Figure 4 jof-11-00459-f004:**
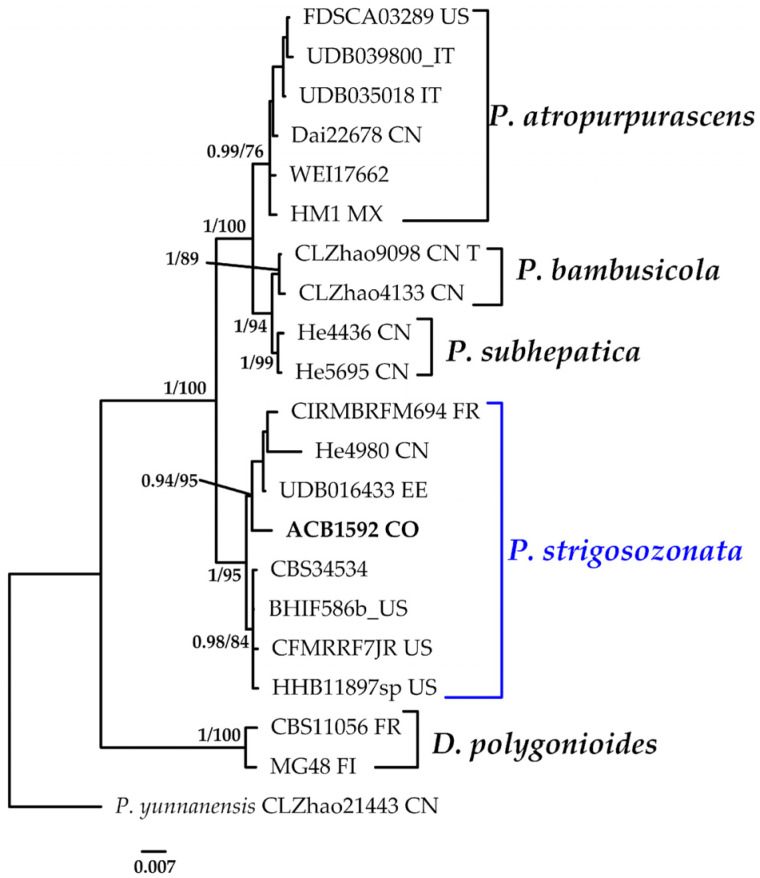
Phylogenetic relationships within the *Punctularia* clade inferred from a combined dataset of ITS and nLSU sequences using BI. Specimens sequenced in this study are indicated in bold. Values at nodes represent Bayesian Posterior Probabilities (BPPs, left) and Shimoradai–Hasegawa approximate likelihood ratio test (SH, right). Type specimens are marked with a (T). Country codes following voucher specimens correspond to their country of origin. The bar indicates the expected number of substitutions per site.

**Figure 5 jof-11-00459-f005:**
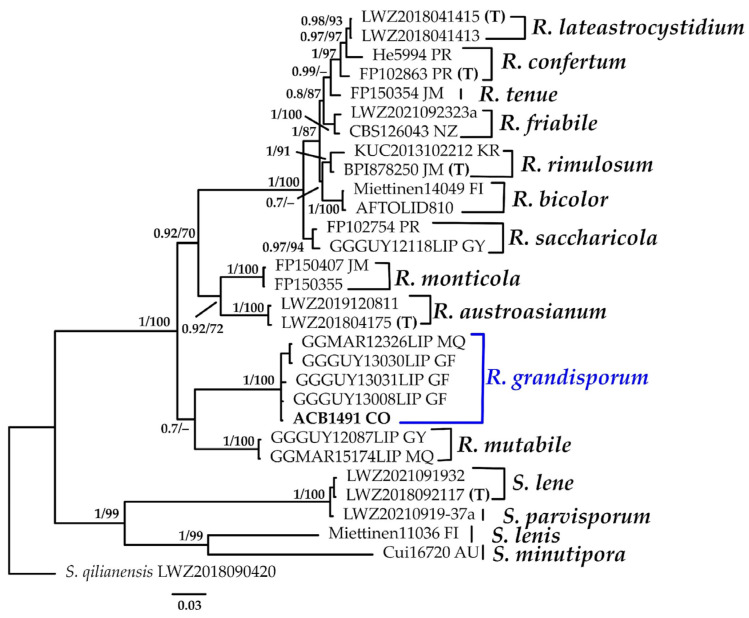
Phylogenetic relationships within the *Resinicium *clade inferred from a combined dataset of ITS and nLSU sequences using BI. Specimens sequenced in this study are indicated in bold. Values at nodes represent Bayesian Posterior Probabilities (BPPs, left) and Shimoradai–Hasegawa approximate likelihood ratio test (SH, right). A minus sign (−) indicates values lower than 70%. Type specimens are marked with a (T). Country codes following voucher specimens correspond to their country of origin. The bar indicates the expected number of substitutions per site.

**Figure 6 jof-11-00459-f006:**
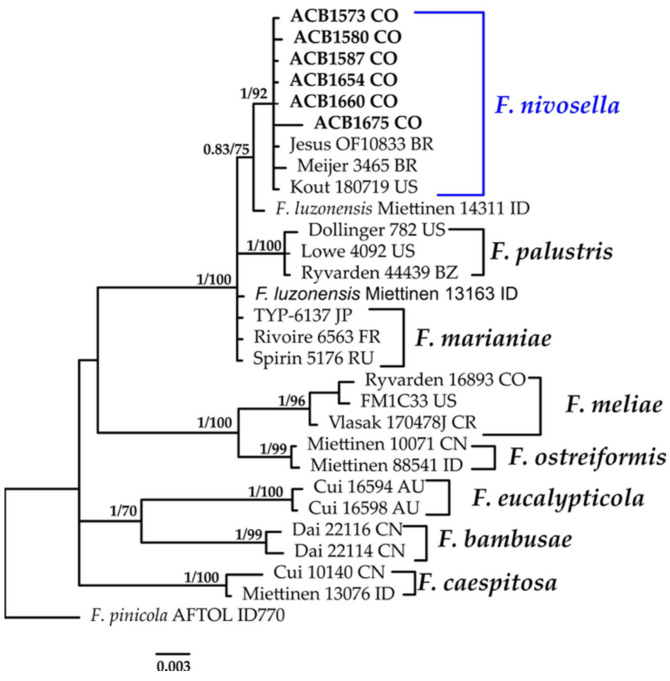
Phylogenetic relationships within the *Fomitopsis *(*Pilatoporus* group) clade inferred from a combined dataset of ITS and nLSU sequences using BI. Specimens sequenced in this study are indicated in bold. Values at nodes represent Bayesian Posterior Probabilities (BPPs, left) and Shimoradai–Hasegawa approximate likelihood ratio test (SH, right). Country codes following voucher specimens correspond to their country of origin. The bar indicates the expected number of substitutions per site.

**Figure 7 jof-11-00459-f007:**
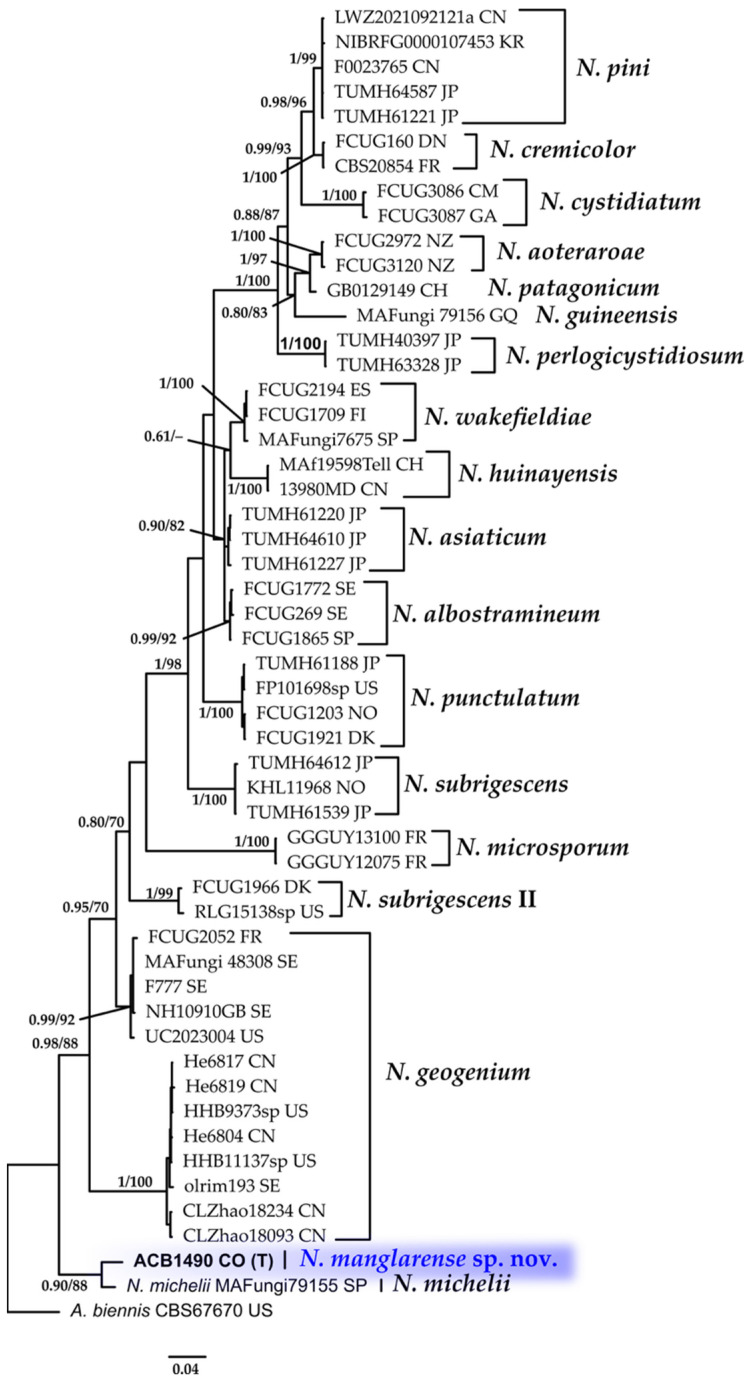
Phylogenetic relationships within the *Neohypochnicium *clade inferred from a combined dataset of ITS and nLSU sequences using BI. Specimens sequenced in this study are indicated in bold. Values at nodes represent Bayesian Posterior Probabilities (BPPs, left) and Shimoradai–Hasegawa approximate likelihood ratio test (SH, right). A minus sign (−) indicates values lower than 70%. Type specimens are marked with a (T). Country codes following voucher specimens correspond to their country of origin. The bar indicates the expected number of substitutions per site.

**Figure 8 jof-11-00459-f008:**
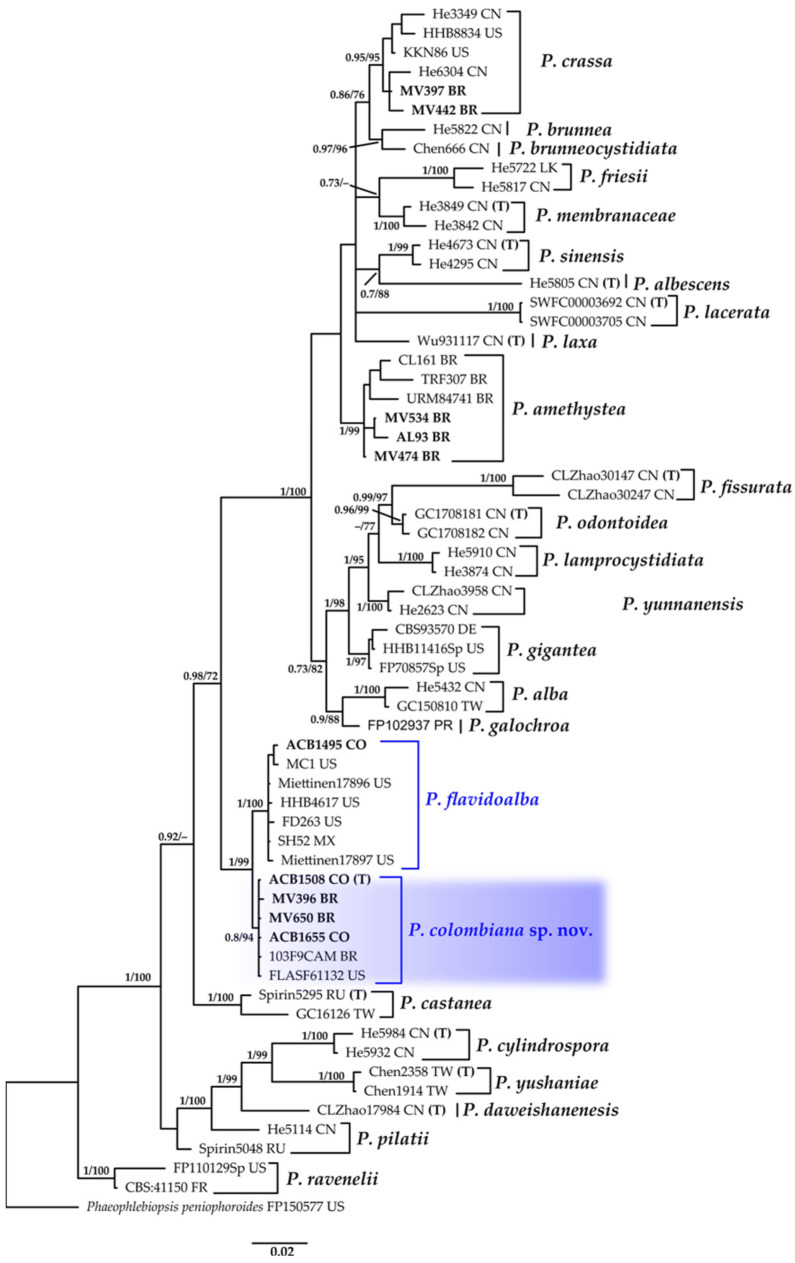
Phylogenetic relationships within the *Phlebiopsis *clade inferred from a combined dataset of ITS and nLSU sequences using BI. Specimens sequenced in this study are indicated in bold. Values at nodes represent Bayesian Posterior Probabilities (BPPs, left) and Shimoradai–Hasegawa approximate likelihood ratio test (SH, right). A minus sign (−) indicates values lower than 70%. Type specimens are marked with a (T). Country codes following voucher specimens correspond to their country of origin. The bar indicates the expected number of substitutions per site.

**Figure 9 jof-11-00459-f009:**
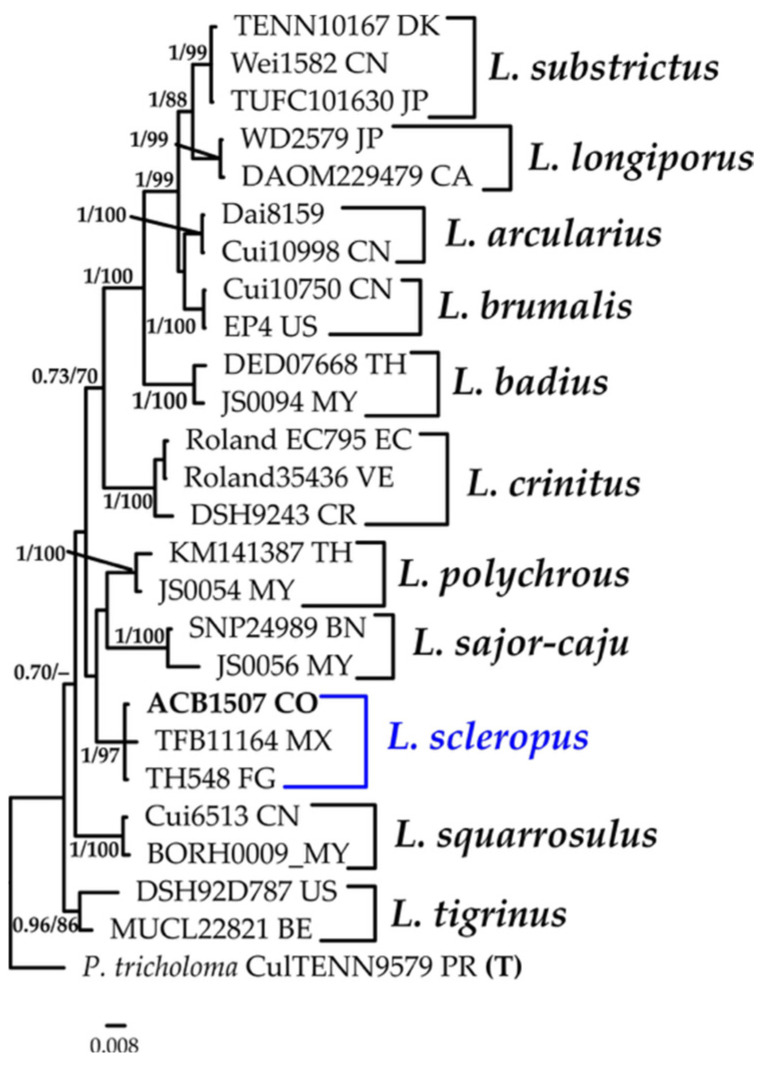
Phylogenetic relationships within the *Lentinus *clade inferred from a combined dataset of ITS and nLSU sequences using BI. Specimens sequenced in this study are indicated in bold. Values at nodes represent Bayesian Posterior Probabilities (BPPs, left) and Shimoradai–Hasegawa approximate likelihood ratio test (SH, right). A minus sign (−) indicates values lower than 70%. Type specimens are marked with a (T). Country codes following voucher specimens correspond to their country of origin. The bar indicates the expected number of substitutions per site.

**Figure 10 jof-11-00459-f010:**
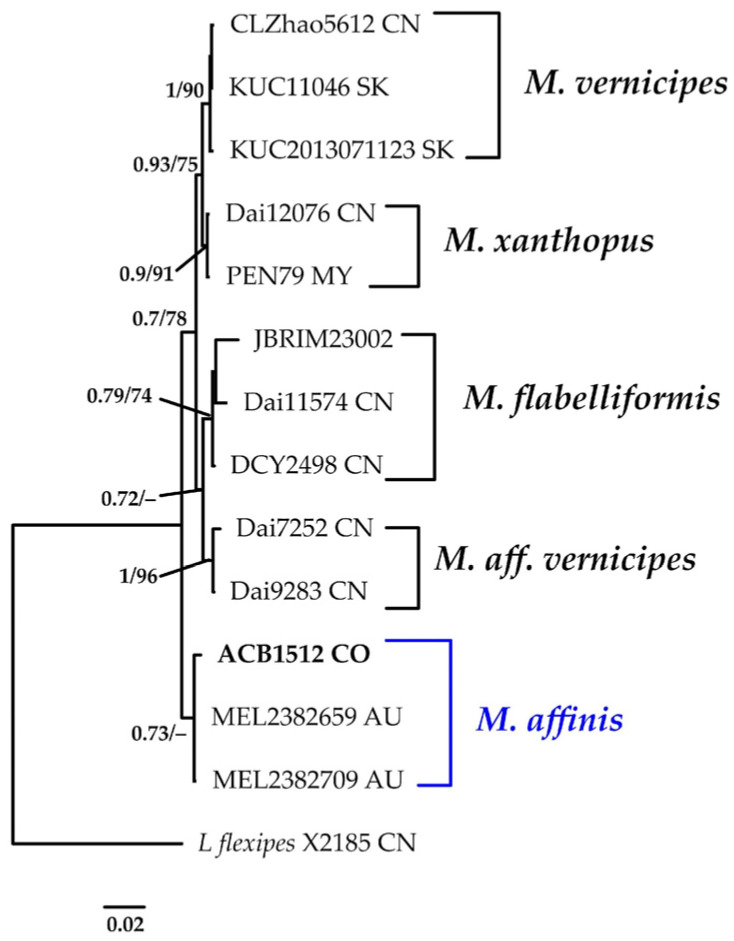
Phylogenetic relationships within the *Microporus *clade inferred from a combined dataset of ITS and nLSU sequences using BI. Specimens sequenced in this study are indicated in bold. Values at nodes represent Bayesian Posterior Probabilities (BPPs, left) and Shimoradai–Hasegawa approximate likelihood ratio test (SH, right). A minus sign (−) indicates values lower than 70%. Country codes following voucher specimens correspond to their country of origin. The bar indicates the expected number of substitutions per site.

**Figure 11 jof-11-00459-f011:**
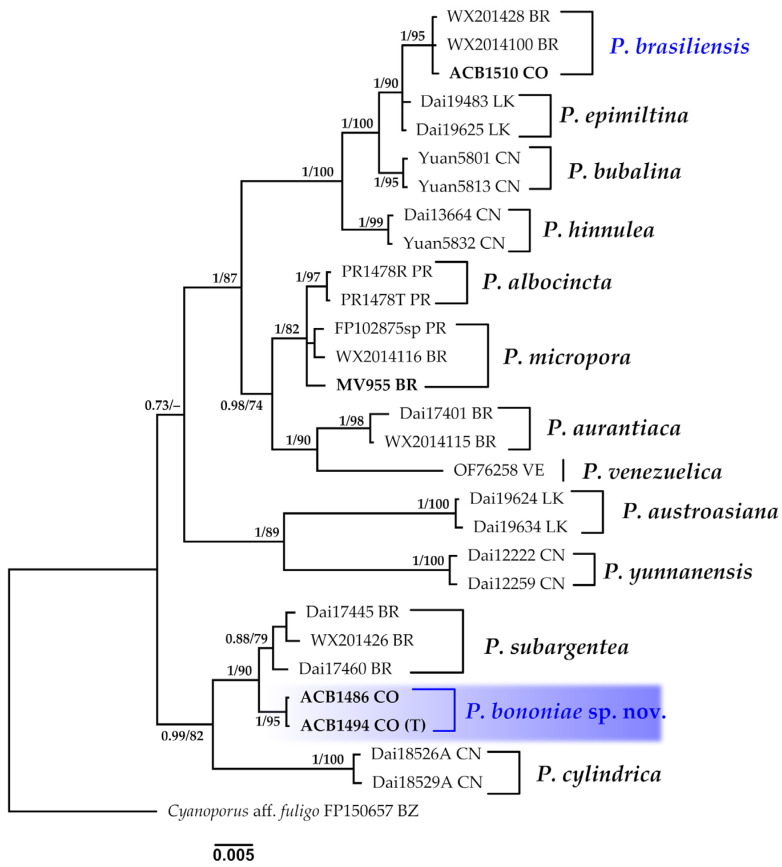
Phylogenetic relationships within the *Porogramme *clade inferred from a combined dataset of ITS and nLSU sequences using BI. Specimens sequenced in this study are indicated in bold. Values at nodes represent Bayesian Posterior Probabilities (BPPs, left) and Shimoradai–Hasegawa approximate likelihood ratio test (SH, right). A minus sign (−) indicates values lower than 70%. Country codes following voucher specimens correspond to their country of origin. The bar indicates the expected number of substitutions per site.

**Figure 12 jof-11-00459-f012:**
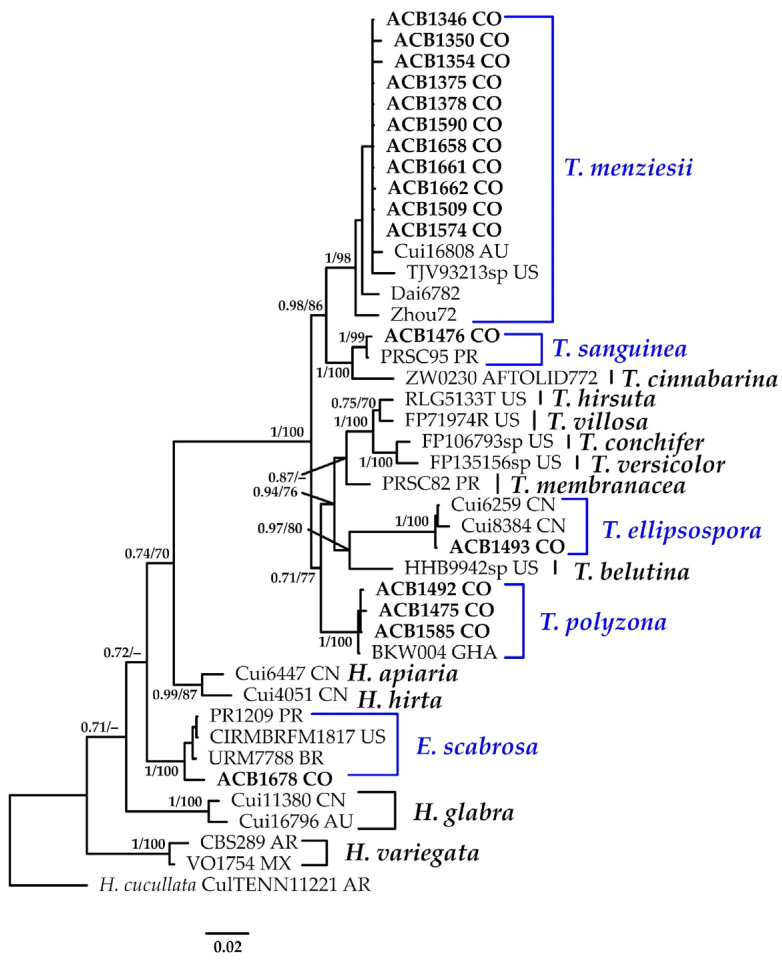
Phylogenetic relationships within the *Trametes *clade inferred from a combined dataset of ITS and nLSU sequences using BI. Specimens sequenced in this study are indicated in bold. Values at nodes represent Bayesian Posterior Probabilities (BPPs, left) and Shimoradai–Hasegawa approximate likelihood ratio test (SH, right). A minus sign (−) indicates values lower than 70%. Country codes following voucher specimens correspond to their country of origin. The bar indicates the expected number of substitutions per site.

**Figure 13 jof-11-00459-f013:**
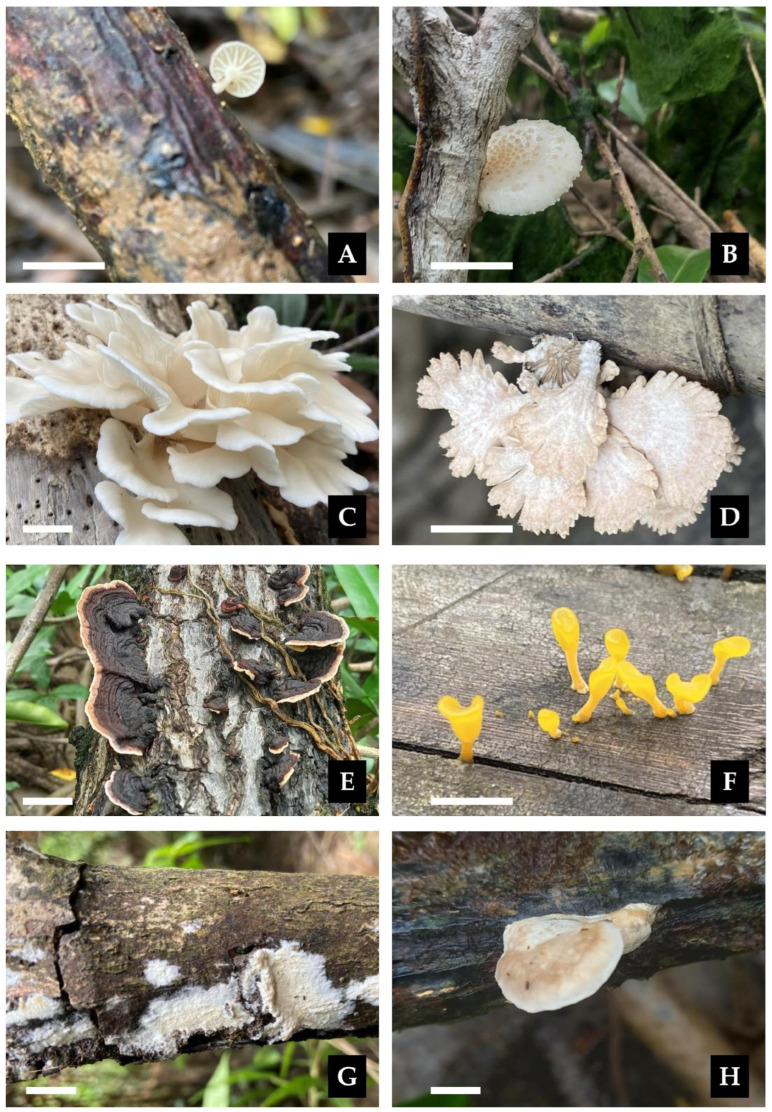
Fresh basidiomes of diverse macrofungi in mangroves of the Colombian Pacific. (**A**) *P. palomivorus*. (**B**) *O. platensis*. (**C**) *P. djamor*. (**D**) *S. commune*. (**E**) *P. strigosozonata*. (**F**) *D. spathularia*. (**G**) *R. grandisporum*. (**H**) *F. nivosella*. Scale bars: 5 mm. Photos by Bolaños-Rojas, A.C.

**Figure 14 jof-11-00459-f014:**
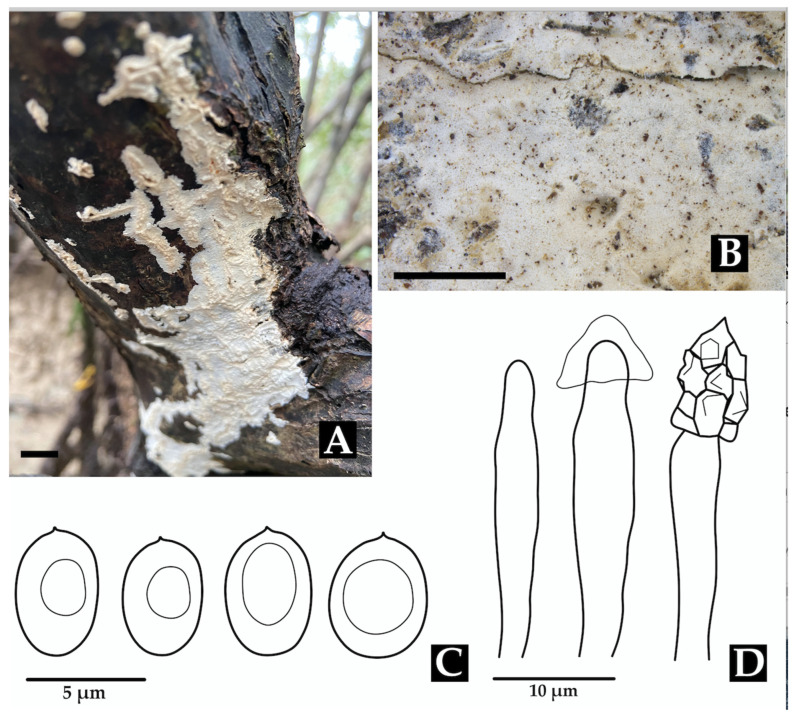
Fresh basidiome of *N. manglarense* sp. nov. (ACB1490, holotype). (**A**) Macroscopic basidiome on decaying wood of *R. mangle*. (**B**) Hymenophore. (**C**) Basidiospores. (**D**) Cystidia. Scale bar: A = 1 cm, B = 2 mm. Photos by Bolaños-Rojas, A.C. Drawings by Motato-Vásquez, V.

**Figure 15 jof-11-00459-f015:**
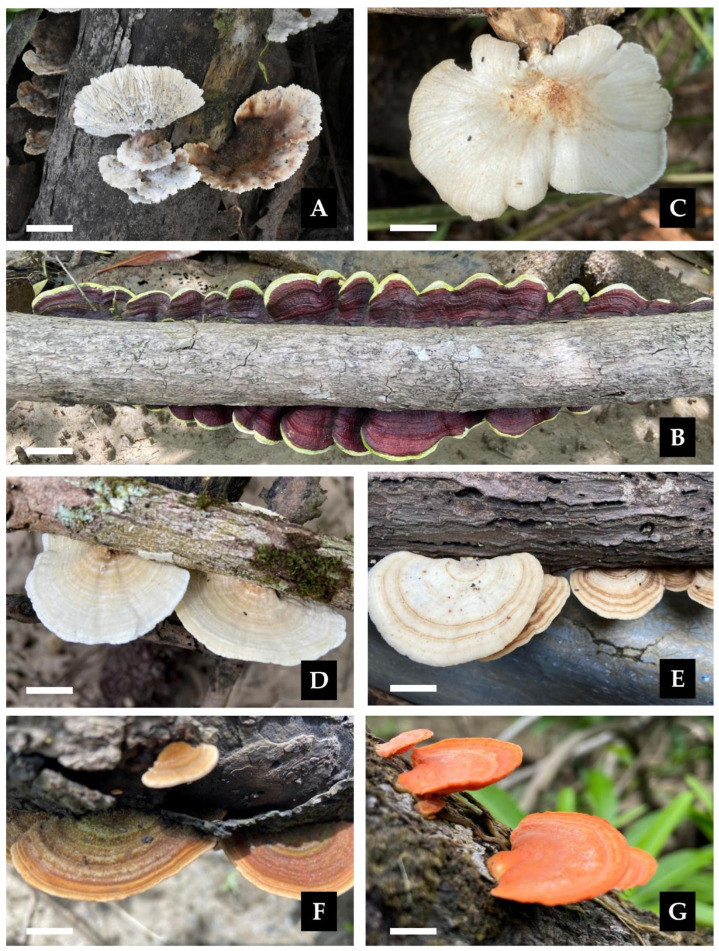
Fresh basidiomata of diverse macrofungi in mangroves of the Colombian Pacific. (**A**) *C. dendriticum*. (**B**) *E. scabrosa*. (**C**) *L. scleropus*. (**D**) *T. ellipsospora*. (**E**) *T. menziesii*. (**F**) *T. polyzon*a. (**G**) *T. sanguineus*. Scale bars: 5 mm. Photos by Bolaños-Rojas, A.C.

**Figure 16 jof-11-00459-f016:**
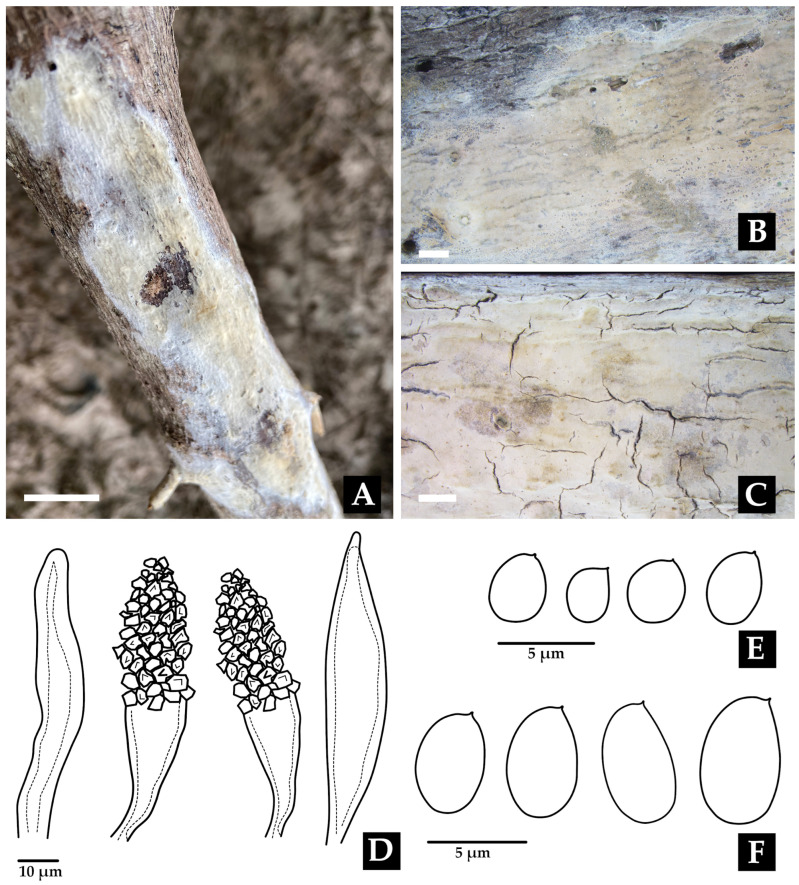
Fresh basidiomes of *Phlebiopsis* species from mangrove. (**A**) Fresh basidiome of *P. flavidoalba* (ACB1495). (**B**) Hymenophore of *P. flavidoalba* (ACB1495). (**C**) Fresh basidiome of *P. colombiana *(ACB1508, holotype). (**D**) Lamprocystidia of *P. colombiana *(ACB1508, holotype). (**E**) Basidiospores of *P. colombiana *(ACB1508, holotype). (**F**) Basidiospores of *P. flavidoalba* (ACB1495). Scale bar: **A** = 1 cm, **B** = 2 mm. Photos by Bolaños-Rojas, A.C. Drawings by Motato-Vásquez, V.

**Figure 17 jof-11-00459-f017:**
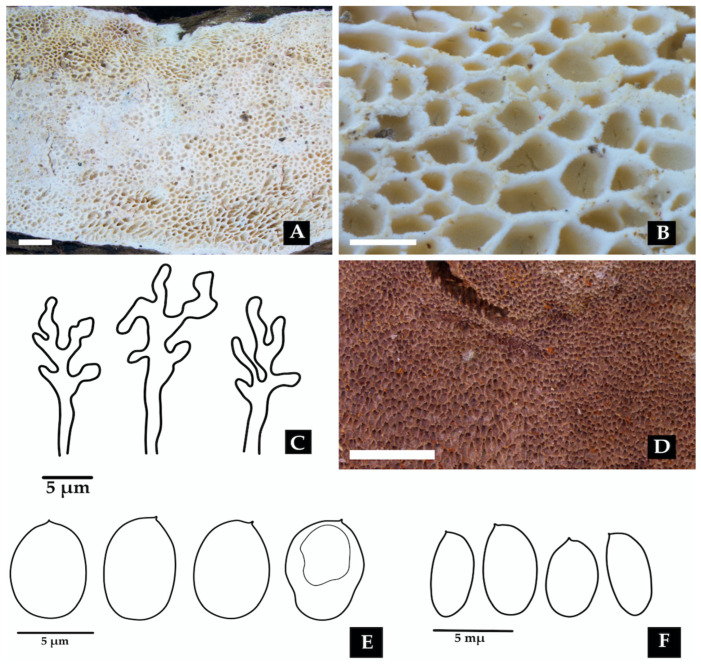
Fresh basidiomes of *Porogramme* species in the Colombian Pacific mangrove. (**A**) Fresh basidiome of *P. bononiae* sp. nov. (ACB1494, holotype). (**B**) Hymenophore (ACB1494). (**C**) Dendrohyphidias (ACB1494). (**D**) Hymenophore of *P. brasiliensis* (ACB1510). (**E**) Basidiospores of *P. bononiae* sp. nov. (ACB1494, holotype). (**F**) Basidiospores of *P. brasiliensis *(ACB1510). Scale bars: **A** = 2 mm, **B** = 0.5 mm, and **D** = 2 mm. Photograms by Bolaños-Rojas, A.C. Drawings by Motato-Vásquez.

**Table 1 jof-11-00459-t001:** List of sequences generated in this study. For each collection, the species name, voucher, locality, and GenBank accession number are provided. Type specimens are denoted with an asterisk (*). Missing information is indicated with an en dash (–). New species sequences are indicated in bold.

Specimen	Voucher	Locality	GenBank Accession No.	
			ITS	nLSU
*F. nivosella* (Murrill) Spirin & Vlasák	ACB1573	CO	PV330456	–
	ACB1580	CO	PV330457	PV299508
	ACB1587	CO	PV330458	PV299509
	ACB1654	CO	PV330459	PV299510
	ACB1660	CO	PV330460	PV299511
	ACB1675	CO	PV330461	PV299512
*Lentinus scleropus* (Pers.) Fr.	ACB1507	CO	PV330462	PV299513
*M. affinis* (Blume & T. Nees) Kuntze	ACB1512	CO	PV330463	PV299514
***N. manglarense* sp. nov.**	**ACB1490 ***	**CO**	**PV330464**	**–**
*O. platensis* (Speg.) Speg.	ACB1478	CO	PV330465	PV299515
	ACB1571	CO	PV330467	PV299517
*Paramarasmius palmivorus* (Sharples) Antonín & Kolarík	ACB1519	CO	PV330467	PV299517
*P. amethystea* (Hjortstam & Ryvarden) R.S. Chikowski & C.R.S. de Lira	MV534	BR	PV562813	PV562973
	AL93	BR	PV562814	PV562974
	MV474	BR	PV562815	PV562975
***P. colombiana*** **sp. nov.**	**ACB1508 ***	**CO**	**PV330468**	**PV299518**
	**MV396**	**BR**	**PV562816**	**PV562976**
	**MV650**	**CO**	**PV562817**	**–**
	**ACB1655**	**CO**	**PV562818**	**–**
*P. crassa* (Lév.) Floudas y Hibbett	MV 397	BR	PV562819	PV562977
	MV 442	BR	PV562820	–
*P. flavidoalba* (Cooke) Hjortstam	ACB1495	CO	PV330469	PV299519
***P. bononiae* sp. nov.**	**ACB1486**	**CO**	**PV330470**	**PV299520**
	**ACB1494 ***	**CO**	**PV330471**	**PV299521**
*P. brasiliensis* (Ryvarden) Y.C. Dai, W.L. Mao & Yuan Yuan	ACB1510	CO	PV330472	PV299522
*P. micropora* (A.M.S. Soares & W.K.S. Waxier) Y.C. Dai, W.L. Mao & Yuan Yuan	MV955	BR	PV562821	PV562978
*P. strigosozonata* (Schwein.) P.H. Talbot	ACB 1592	CO	PV330473	PV299523
*Resinicium grandisporum* G. Gruhn, Dumez & Schimann	ACB1491	CO	PV330475	PV299525
*Earliella scabrosa* (Pers.) Gilb. & Ryvarden	ACB1678	CO	PV330455	PV299507
*T. ellipsospora* Ryvarden	ACB1493	CO	PV330477	PV299528
*T. menziesii* (Berk.) Ryvarden	ACB1350	CO	PV330478	PV299529
	ACB1354	CO	PV330479	PV299530
	ACB1375	CO	PV330480	PV299531
	ACB1378	CO	PV330481	PV299532
	ACB1574	CO	PV330482	PV299533
	ACB1590	CO	PV330483	PV299534
	ACB1658	CO	PV330484	PV299535
	ACB1661	CO	PV330485	PV299536
	ACB1662	CO	PV330486	PV299537
*T. polyzona* (Pers.) Justo	ACB1475	CO	PV330487	PV299538
	ACB1492	CO	PV330488	PV299539
	ACB1585	CO	PV330489	PV299540
*T. sanguinea* (Klotzsch) Pat.	ACB1476	CO	PV330474	PV299524

All accession numbers utilized for phylogenetic analysis are available in the Appendix A.

## Data Availability

All sequences have been deposited in GenBank (https://www.ncbi.nlm.nih.gov, accessed on 16 February 2025) and MycoBank (https://www.mycobank.org, accessed on 20 February 2025). The R codes are available upon request.

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
