# Peer review of "Hidden Treasures of Colombia’s Pacific Mangrove: New Fungal Species and Records of Macrofungi (Basidiomycota)"

_jof, 2025, doi:10.3390/jof11060459_

Round 1

Reviewer 1 Report

This paper presents the first survey of macrofungi from mangrove ecosystems in Colombia and introduces three new species. which is both interesting and valuable for understanding the biodiversity within different ecosystems. The text is well prepared, but some minor adjustments are needed.

For instance,

Line 21: 'basidiomes', Please confirm whether these are basidiomes or samples?

Line 103 and 106: change 'Sporomes' to basidiocarps or basidiomes.

Line 169: in the table, a column of references needs to be added in the table.

....

Line 1181: in the discussion section, it is recommended to be more succinct, eschew redundant expressions, and emphasize the core viewpoints.

Some comments were provided directly on the manuscript.

Reviewer 2 Report

The paper entitled 'Hidden Treasures of Colombia's Pacific Mangrove: New Fungal Species and Records of Macrofungi (Basidiomycota)' is devoted to study of Basidiomycota diversity in Mangrove in Colombia. This work looks timely, because mangrove biome, espcially fungal, is weakly investigated. The authors performed complex investigation of macrofungal diversity using morphological and molecular methods, and identified three new species; eight species were identified in Colombia for the first time. I believe th paper is scientifically significant and can be accepted for publication. At the same time, I have some notes to slightly improve the manuscript.

  1. The major part of the 'Results' section is the description of phylogeny of the identified fungal genus. Each subsection includes phylogenetic tree and description of them and their phylogenetic properties. I propose to aggregate this information into a table and move the dendrogramms (or some of them) to Supplementary. Probably, this format would demonstrate the data more clearly.
  2.   I also suggest to make separate section for more detailed description of novel species and combine phylogenetic +morphological results in these sections.  

1. Section 2.1., Lines 90-97: How many samples were analyzed totally?

2. Section 2.2., Lines 123-128: References for primers used should be provided.

3. The authors present phylogenetic data only for concatenated sequence ITS+nLSU. It would be also interesting to estimate phylogenetic properties for each of these barcodes separately.
